# Analysis of ENSO-Driven Variability, and Long-Term Changes, of Extreme Precipitation Indices in Colombia, Using the Satellite Rainfall Estimates CHIRPS

**Juan Diego Giraldo-Osorio** [1,*,†] **, David Enrique Trujillo-Osorio** [1,†] **and Oscar Manuel Baez-Villanueva** [2,3,†]

1    Research Group Ciencia e Ingeniería del Agua y el Ambiente, Civil Enginnering Deparment, School of Enginnering, Pontificia Universidad Javeriana, Bogotá 110231, Colombia; david.trujillo@javeriana.edu.co
2    Institute for Technology and Resources Management in the Tropics and Subtropics (ITT), Technology Arts Sciences TH Köln, 50679 Cologne, Germany; oscar_manuel.baez_villanueva@smail.th-koeln.de
3    Faculty of Spatial Planning, TU Dortmund University, 44221 Dortmund, Germany
*    Correspondence: j.giraldoo@javeriana.edu.co; Tel.: +57-1-320-8320 (ext. 5350)
†    These authors contributed equally to this work.

**Abstract:** Climate change includes the change of the long-term average values and the change of the tails of probability density functions, where the extreme events are located. However, obtaining average values are more straightforward than the high temporal resolution information necessary to catch the extreme events on those tails. Such information is difficult to get in areas lacking sufficient rain stations. Thanks to the development of Satellite Precipitation Estimates with a daily resolution, this problem has been overcome, so Extreme Precipitation Indices (EPI) can be calculated for the entire Colombian territory. However, Colombia is strongly affected by the ENSO (El Niño—Southern Oscillation) phenomenon. Therefore, it is pertinent to ask if the EPI's long-term change due to climate change is more critical than the anomalies due to climate variability induced by the warm and cold phases of ENSO (El Niño and La Niña, respectively). In this work, we built EPI annual time series at each grid-point of the selected Satellite Precipitation Estimate (CHIRPSv2) over Colombia to answer the previous question. Then, the Mann-Whitney-Wilcoxon test was used to compare the samples drawn in each case (i.e., change tests due to both long-term and climatic variability). After performing the analyses, we realized that the importance of the change depends on the region analyzed and the considered EPI. However, some general conclusions became evident: during El Niño years (La Niña), EPI's anomaly follows the general trend of reduction -drier conditions- (increase; -wetter conditions-) observed in Colombian annual precipitation amount, but only on the Pacific, the Caribbean, and the Andean region. In the Eastern plains of Colombia (Orinoquía and Amazonian region), EPI show a certain insensitivity to change due to climatic variability. On the other hand, EPI's long-term changes in the Pacific, the Caribbean, and the Andean region are spatially scattered. Still, long-term changes in the eastern plains have a moderate spatial consistency with statistical significance.

**Keywords:** Colombia; Extreme Precipitation Index (EPI); long-term change; climate change; climate variability; Satellite Precipitation Estimate (SPE); CHIRPS; Mann-Whitney-Wilcoxon test; trend analysis

## 1. Introduction

Climate change is related to the change in the long-term mean of precipitation and temperature values. However, urge to understand the change in the extreme events, because these climate phenomena produce uncountable damages to the human life and extensive impacts on the ecosystems [1]. Various studies have shown that the occurrence of some extreme events (e.g., extreme precipitation events and heat waves) have had a human contribution [2–5]. According to [4], the moderate precipitation events have increased by 18% due to the Earth's temperature rise since the pre-industrial era. In this sense, climate

models predict a continuous increase of temperature, which is related to an increase in extreme precipitation events during the 21st and in some regions an intensification of the water cycle [5–9].

Generally, studies that work with climatic extremes use daily or sub-daily precipitation and temperature data, either from gauge stations, Reanalysis data, or climate models simulations [5,10,11]. However, sometimes the analysis of ground-based precipitation data suggests a reduction in extreme precipitation events in regions where temperatures are increasing [12]. These results generate uncertainty about the future trend of extreme precipitation events because, in general, a worsening of extreme events is expected with a positive change of mean temperatures [13].

Most of the studies that analyse climate change scenarios are focused on annual averages. However, the inclusion of daily data, although more difficult to process, provide insights related to the extreme precipitation events, which are responsible for a large part of the environmental and social damage attributed to the climate [14,15]. Despite their importance, these observations are less frequent and, thus, more difficult to acquire. In this sense, data-scarce regions, such as Colombia, lack of dense networks of meteorological stations, which hinders the spatio-temporal characterisation of precipitation [16]. However, several precipitation products (*P* products) provide estimates related to the spatio-temporal distribution of precipitation, which enables the analysis of precipitation extremes over data-scarce regions [17–23].

The El Niño—Southern Oscillation (ENSO) has a strong influence on the global climate system [24,25] as it is one of the main drivers of the inter-annual variability of precipitation extremes in different regions of the planet (i.e., droughts spells and heavy rain events). Therefore, understanding the influence that the ENSO has on the extreme precipitation events is pivotal to assess their economic impacts and the risks associated with them to strive towards the implementation of adaptation measures [26,27]. These measures are of utmost importance for Colombia since the country is frequently affected by extreme ENSO events, which differ depending on the ENSO phase (i.e., El Niño/La Niña). For example, in La Niña event of 2010–2011, heavy precipitation, floods, and landslides affected four million people in Colombia and caused a financial loss of about US$8 billion [28]. On the other hand, severe reductions in precipitation and river flows have been widely recorded during El Niño years in Colombia [28–31], which have led to energy cutoffs due to the great magnitude of the droughts that frequently impact the country's economy [32–34].

The majority of studies that focus on extreme events select only one event per year, following the classic hydrology's definition of return period [15]. This approach assumes that the extracted annual values are considered a set of *independent* and *identically distributed* random variables (*i.i.d.*); therefore, a probability distribution could be used to fit the data. However, sometimes the water processes are affected by inter-annual variability (i.e., periods greater than one year), causing that the annual values can not be considered a set of *i.i.d.* random variables. In this sense, the extreme events may belong to different populations. In our study, we used the ENSO phases distinguish between those populations of extreme events. The ENSO has two extreme phases with a quasi-period occurrence from 3 to 7 years: (i) El Niño (the warm phase) and (ii) La Niña (the cold phase), in addition to the average conditions (normal years) [24].

This manuscript wants to answer the question: ¿Is the EPIs long-term change more critical than the EPIs anomalies driven by the ENSO phenomenon? Then, the main objective of the present work is to analyze diverse Extreme Precipitation Indices (EPI) over Colombia from two viewpoints: (i) the long-term change of the EPIs and (ii) the inter-annual variability of the EPIs considering the ENSO phases. However, the Colombian territory is large and diverse, and then another scientific question arises: ¿is the EPIs behavior the same for all the study region? The spatial analysis locates those areas where the long-term changes (or the inter-annual anomalies) are the most important, and link this behavior with their specific climate characteristics.

## 2. Study Area

Colombia is a Latin-American country located in the northwest region of South America. It has access to both the Pacific and Atlantic Ocean (through the Caribbean Sea). It is bounded to the east with Venezuela, to the south with Brazil, Peru, and Ecuador, and the northwest with Panama (Figure 1a).

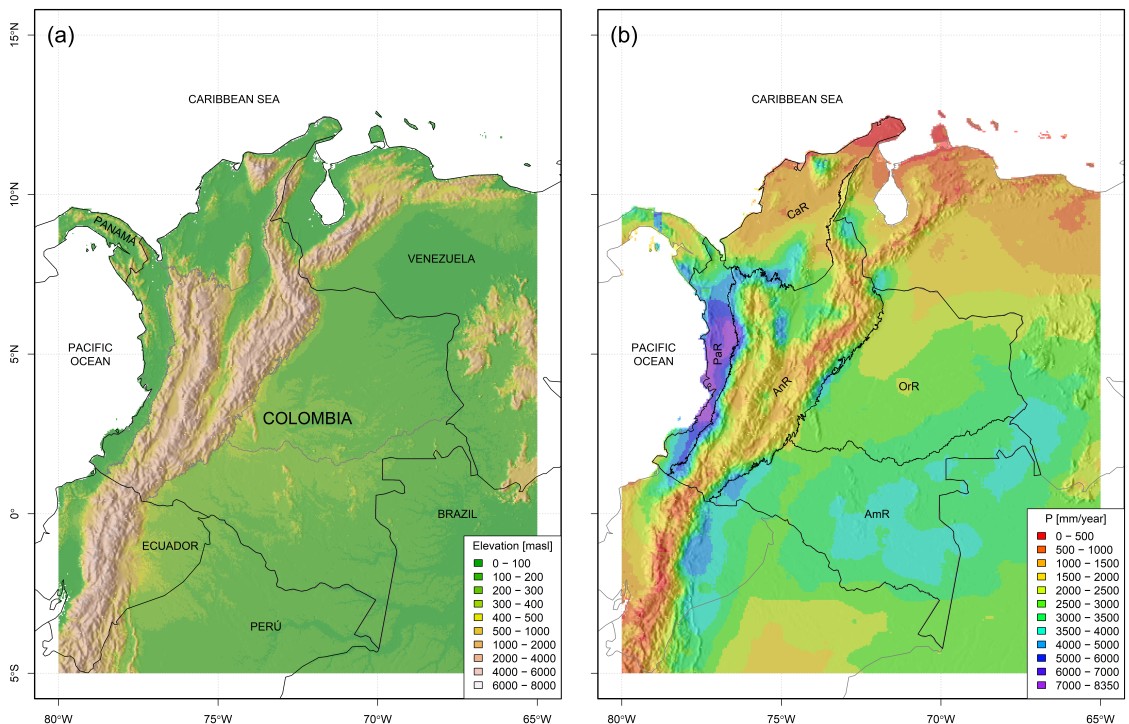

**Figure 1.** Location of the study area. The maps correspond to (**a**) the elevation of Colombia and (**b**) the mean annual precipitation. Also in pannel (**b**), the continental natural regions of Colombia are depicted: the Caribbean region (CaR); the Pacific region (PaR); the Andean region (AnR); the Orinoco region (OrR); and the Amazon region (AmR).

Colombia's climate is fueled by moisture inputs from both oceans and the Amazon rainforest to the southeast. Nevertheless, its spatial distribution of precipitation is strongly driven by the Andean Mountain Chain, which crosses the territory in the Southwest-Northeast direction (Figure 1a,b) [35]. In this way, the continental territory of Colombia can be divided into five natural regions ([36,37]), which share some climatic features: (i) the Caribbean region (CaR), which is the driest and northernmost region of Colombia; (ii) the Pacific region (PaR), which is a super-humid narrow area close to the Pacific coast in the west; (iii) the Andean region (AnR), which is the central mountainous area that the Andes Mountain Chain crosses; (iv) the Orinoco region (OrR), which is the eastern plain dry savanna of Orinoco River Basin; and (v) the Amazon region (AmR), which covers the humid river basin of the Amazonas River.

The maps of mean seasonal precipitation of the study area (Figure 2) reveal particular behaviours: the AnR has two precipitation peaks (MAM and SON), which coincide with the Intertropical Convergence Zone (ITCZ) passage over the mountain chains; the OrR and the AmR have their precipitation peak on JJA; the CaR has the peak during SON, while the PaR has high precipitation values in all seasons. All regions have the lowest precipitation values on DJF. The previous distinction of the annual precipitation cycle between natural regions is fundamental because that condition influences all EPI computation, especially for those EPI related to both dry and wet spells.

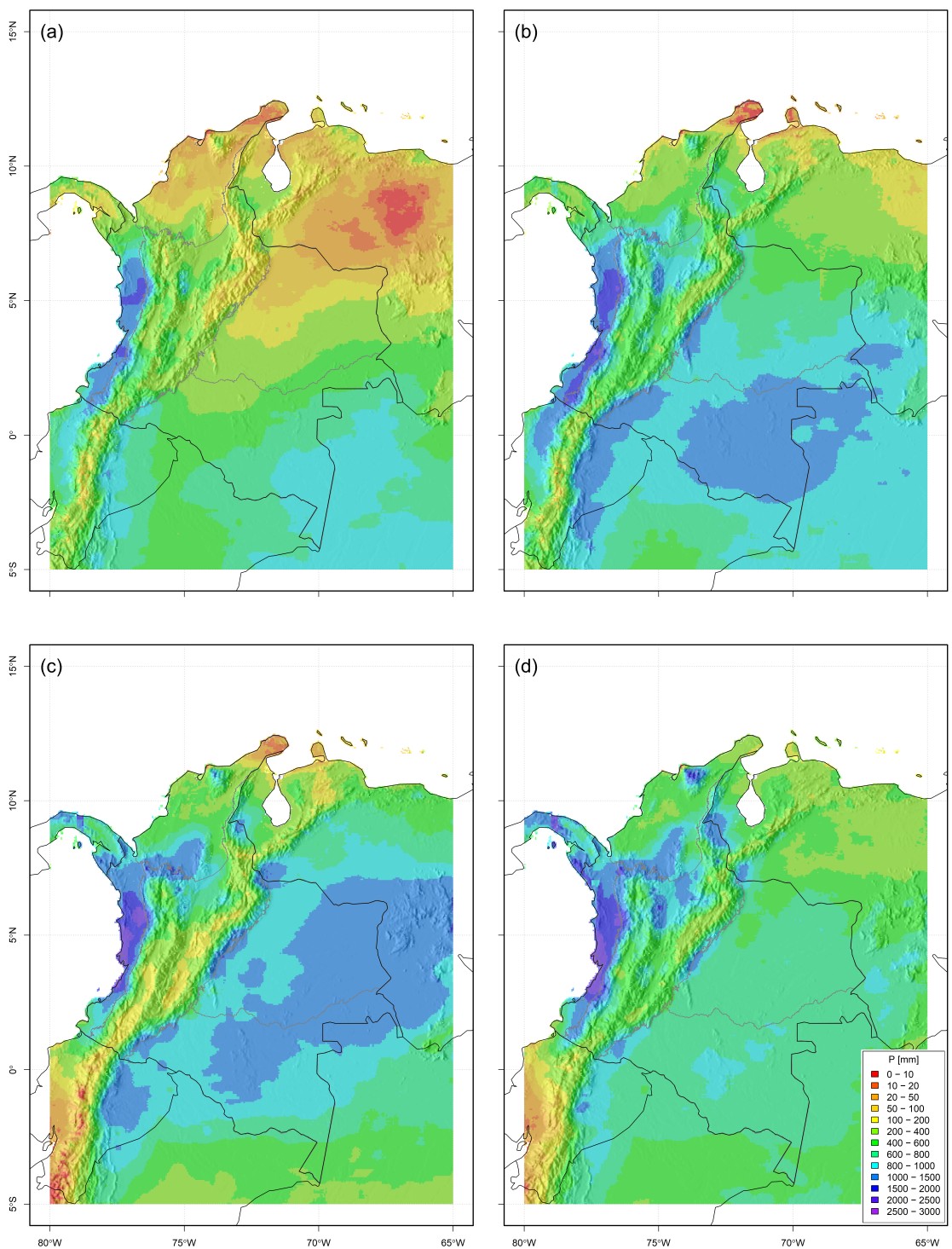

**Figure 2.** Mean seasonal precipitation in the study area computed for (**a**) the northern hemisphere winter (DJF); (**b**) spring (MAM); (**c**) summer (JJA); and (**d**) autumn (SON). These figures were developed using CHIRPSv2 dataset (see Section 3).

## 3. Data

The Climate Hazards Group InfraRed Precipitation with Station data version 2 (CHIRPSv2; [22]) is a 0.05° daily precipitation product with a quasi-global coverage (50° N–50° S). It was designed to monitor environmental changes over land and agricultural drought. CHIRPSv2 uses the Tropical Rainfall Measuring Mission Multi-satellite Precip-

itation Analysis version 7 (TRMM 3B42v7) to calibrate the global cold cloud duration rainfall estimates while using a monthly station data merging approach to reduce their precipitation estimates bias [22]. CHIRPSv2 was selected because (i) of its relative long temporal coverage (starting in 1981); (ii) of its relatively high spatial resolution (0.05°); and (iii) it has provided good results when evaluated over Colombia [38–40] Particularly, Báez-Villanueva et al., (2018) [39] showed that, at daily and monthly scales, CHIRPSv2 performed the best over the Magdalena river basin (the most important basin in Colombia). Also, CHIRPS data are suitable for our study because the IDEAM's meteorology office has used and evaluated the CHIRPS data in their activities since the product release in 2015 (IDEAM is *Instituto de Hidrología, Meteorología y Estudios Ambientales*; a government agency that studies the weather and climate of Colombia, among other subjects):

- In Funk et al., (2015) [22], 308 IDEAM's stations were used to validate the CHIRPS product in Colombia, an area with complex topography where tropical processes mainly drive the rain.
- Since that, IDEAM has tested the performance of CHIRPS using precipitation data from its stations network. For example, the technical report IDEAM–METEO/002-2018 [41] compared about a thousand precipitation stations' records with CHIRPS data. They conclude that CHIRPS data are well correlated with the station data. However, there are still some problems with the product bias.
- IDEAM uses CHIRPS to build monthly maps of precipitation in Colombia and other precipitation products (e.g., the monthly precipitation anomaly). Those monthly maps support the forecasting models used in the agency (technical report IDEAM–METEO/001-2020 [42]).

## 4. Methodology

The study goal is to evaluate the long-term change of EPIs and the anomalies of these series during the ENSO extreme phases. For that, the methodology in Figure 3 was followed, which can be summarized as follows:

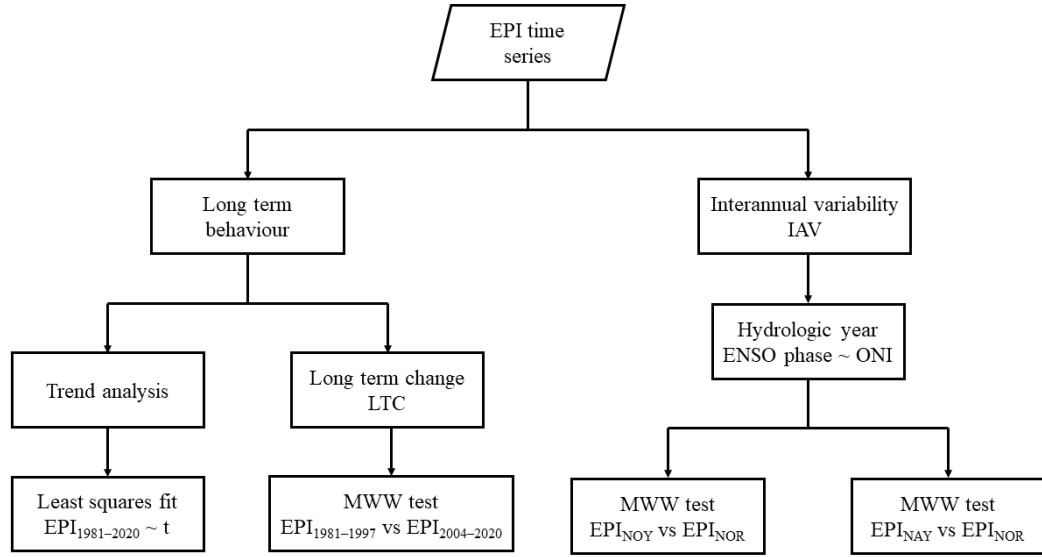

**Figure 3.** Conceptual figure of the methodology followed in this study.

- Construction of the time series of the indices in each grid-cell of CHIRPSv2 in the study area.
- Analysis of the long-term behavior of the EPIs: two statistical tests were performed to analyze the long-term behavior of the EPI series (linear regression and Mann-Whitney-Wilcoxon test).

- Interannual variability: The Mann-Whitney-Wilcoxon test was used to analyze the anomalies during El Niño and La Niña years, against normal (or neutral) years.

### 4.1. Time Series of EPI

The CCI/CLIVAR/JCOMM ETCCDI group developed a set of 27 extreme climate indices, which must be computed usign daily data of precipitation and temperature (defined in Zhang et al., 2011 [15]). These indices have been widely used for assessing the observed trend of extreme climate events of precipitation and temperature around the globe. The extreme climate indices are versatile because they can be computed using data from meteorological stations [43–46], gridded historical data [47,48], and recently for forecasting the extreme events behavior in the future using the CMIP6 global models [1]. From that set, we selected the extreme climate indices related to precipitation (henceforth, these selected indices will be named Extreme Precipitation Indices—EPI). These EPIs require daily precipitation data and were computed annually (for every hydrological year) over each grid-cell of CHIRPSv2 within Colombia, following the definitions presented in Table 1 [14,15]. It must be highlighted that we defined the hydrological year for EPIs computation from one-year's June to next-year May (i.e., from 01–06 to 31–05), instead of analyzing them over the calendar years, to match the developing phases of the ENSO phenomenon. The results of these EPIs correspond to the annual time series for 1981–2020 over every grid-cell (i.e., 40 years long EPIs time series). Despite that we used hydrological years, for easiness, the years will be named according to the year they begin. For example, the hydrological year that starts in 1 June 1981 and ends in 31 May 1982 is noted as 1981.

**Table 1.** Extreme Precipitation Indices (EPI) definitions, as they were used in this work. The definitions are based on both Zhang et al., (2011) [15], and ETCCDI webpage [1].

| EPI ID | EPI Name | EPI Definition | Units |
|---|---|---|---|
| Rx1day | Maximum 1-day precipitation amount | Yearly * maximum 1-day precipitation | mm |
| Wet days | Wet days | Yearly * number of days with $p \geq 1$ mm | day |
| SDII | Simple daily intensity index | Total wet-day precipitation (PRCPTOT) to wet days ratio | mm/day |
| PRCPTOT | Total wet-day precipitation | Yearly * total precipitation from wet days | mm |
| R10mm | Number of heavy precipitation days | Yearly * number of days with $p \geq 10$ mm | day |
| CDD | Consecutive dry days | Yearly * maximum dry spell length (consecutive days), where $p < 1$ mm | day |
| CWD | Consecutive wet days | Yearly * maximum wet spell length (consecutive days), where $p \geq 1$ mm | day |
| R95pTOT | Total precipitation on very wet days | Yearly * total precipitation from days where $p \geq R95$ | mm |

[1] http://etccdi.pacificclimate.org/list_27_indices.shtml (accessed on 1 May 2022). * *Hydrological years* were defined as the time frame from June the 1th in year $i$, to May the 31th in year $i + 1$.

### 4.2. Statistical Tests

#### 4.2.1. Hypothesis Test for Regression Slope

A linear model is given by the following equation:

$$y = \beta_0 + \beta_1 \cdot x + e \tag{1}$$

where $\beta_0$ is a constant, $\beta_1$ is the slope, and $e$ is an error term. Also, $x$ is the independent variable, while $y$ is the dependent variable.

Using the ordinary least squares method, the best-fit regression line is gotten, through the minimization of the sum of squared error (SSE):

$$SSE = \sum_{i=1}^{n} e_i^2 = \sum_{i=1}^{n} (y_i - \hat{y}_i)^2 = \sum_{i=1}^{n} (y_i - b_0 - b_1 \cdot x_i)^2 \tag{2}$$

In the Equation (2), $b_0$ and $b_1$ are the best estimators of the lineal model parameters $\beta_0$ and $\beta_1$, and $n$ is the number of observations $(x_i, y_i)$.

It is not only interesting to assess the trend slope magnitude (coefficient $b_1$), but also their statistical significance. The regression *t*-test is applied to test if the slope value $b_1$ is statistically equal (or different) from zero. The *t*-test follows a t distribution with $df = n - 2$:

$$t = \frac{b_1}{s_b} = \frac{b_1}{s_e / \sqrt{\sum_{i=1}^{n}(x_i - \bar{x})^2}} \tag{3}$$

where $b_1$ corresponds to the sample regression coefficient, and $s_e$ is the residual standard error:

$$s_e = \sqrt{\frac{SSE}{n-2}} \tag{4}$$

Finally, the statistical test for $b_1$ ($H_0 : b_1 = 0$; $H_1 : b_1 \neq 0$), allowed to assess the *p*-value of this parameter. The smaller the *p*-value, the stronger the evidence that support $b_1 \neq 0$ [49,50].

#### 4.2.2. Mann-Whitney-Wilcoxon -MWW- Test

The MWW is a nonparametric test where the null hypothesis states that a randomly selected value from one sample $X = \{X_1, X_2, ..., X_{n_x}\}$ is equally likely to be greater than, or lesser than, another randomly selected value from another sample $Y = \{Y_1, Y_2, ..., Y_{n_y}\}$. The MWW test can be used to investigate if the two independent samples $X$ and $Y$ were drawn from populations with the same distribution. The MWW hypotheses are:

- The data from both groups are independent. The size of $X$ is $n_x$, while $Y$ has $n_y$ data (so, $n_x$ and $n_y$ might not be the same, but the sample size is $N = n_x + n_y$).
- The data can be both ordinal variables or continuous variables. This characteristic of the MWW test makes it very suitable for our analyses: the EPIs can be continuous variables (e.g., *Rx1day*, *PRCPTOT*, *R95pTOT*, etc.), or ordinal variables (e.g., *R10mm*, *CDD*, *CWD*, etc.).
- Null Hypothesis $H_0$: The population of both groups $X$ and $Y$ is the same.
- Alternative Hyphothesis $H_1$: The population of both samples is not the same.

The MWW test entails the calculation of the $U$ statistics, whose distribution is known under the null hyphotesis. The elements of both samples $X$ and $Y$ are joined together into a single set, taking care of identify the elements of each one. Then, the joint sample is sorted in increasing order, and ranks are assigned: the smallest value is rank one (1), while the greatest value is rank $N = n_x + n_y$. The sum of the ranks for samples $X$ and $Y$ are $R_x$ and $R_y$, respectively.

$$\begin{aligned} U_x &= R_x - \frac{n_x \cdot (n_x + 1)}{2} \\ U_y &= R_y - \frac{n_y \cdot (n_y + 1)}{2} \\ U &= min(U_x, U_y) \end{aligned} \tag{5}$$

In the event that small samples, the distribution is tabulated. However, if the number of observations in the samples is large enough ($N > 12$), the normal approximation is quite good [49,50].

#### 4.3. Analysis of the EPIs Long Term Behaviour

#### 4.3.1. Trend Analysis

Linear regression using ordinary least squares, was used to detect the main trend of the EPI time series in the whole period 1981–2020. For this purpose, we defined time (*t*) as independent variable, and the correspondent EPI values as the dependent variables. Then, the linear model is the next:

$$\hat{EPI}_t = b_0 + b_1 \cdot t \tag{6}$$

where $\hat{EPI}_t$ is an estimate value of a particular EPI in year $t$ (where $t \; \epsilon \; (1981, ..., 2020)$), and $b_0$, $b_1$ are the regression coefficients.

The slope parameter $b_1$ was the main concern: the positive and negative values of $b_1$ denoted increasing and decreasing trends of EPI, respectively. Also, the *t*-test was used to test the significance of that value.

The linear regression was applied to EPI time series at each grid-cell in the study area. Then, for each EPI, we got trend maps (parameter $b_1$ maps), together with *p*-value maps, obtained from the *t*-test (Section 4.2.1).

### 4.3.2. Long-Term Change -LTC- of EPIs

To evaluate the long-term change, the MWW test was used to analyse whether two samples from different periods taken from EPI time series, belonged to the same population [49]. The first period (period 1) starts in 1981 and ends in 1997, while the second period (period 2) starts in 2004 and ends in 2020. In that way:

- $EPI_{1981-1997} = \{EPI_{1981}, EPI_{1982}, ..., EPI_{1996}, EPI_{1997}\} \Rightarrow n_x = 17$
- $EPI_{2004-2020} = \{EPI_{2004}, EPI_{2005}, ..., EPI_{2019}, EPI_{2020}\} \Rightarrow n_y = 17$

The MWW test was selected because it is a non-parametric statistical test, which is suitable because it does not assume that the samples have any distribution. Also, we assume that the samples have independent observations, which means that the samples are not paired [49,50]. We obtained *p*-values maps for each EPI, but the MWW test does not show the change sign. Then, we built change maps using the mean values in the selected periods (i.e., $\Delta EPI_{LTC} = \overline{EPI}_{2004-2020} - \overline{EPI}_{1981-1997}$).

### 4.4. Analysis of the EPIs Inter-Annual Variability -IAV-

The ENSO is one of the most critical drivers of inter-annual variability of Colombia's climate ([37,51,52]). Although there are a plethora of indices that monitor the ENSO phenomenon, such as the Multivariate ENSO Index (MEI; [53]), the Southern Oscillation Index (SOI; [54,55]), and indices based on sea surface temperature (SST; [56]), we selected the Oceanic Niño Index (ONI; [57]) because it allowed us to differenciate hydrological years as *El Niño* (NOY), *La Niña* (NAY) and *Normal* (NOR) years.

The ONI classifies a particular hydrological year as El Niño/La Niña if the 3-month running mean of SST anomalies in El Niño 3.4 region [58] exceeds $+0.5\,^{\circ}\text{C}/-0.5\,^{\circ}\text{C}$ at least for five months in a row. In this work, we only considered moderate and strong events (i.e., years where the 3-month running mean of SST anomalies exceeds $+1.0\,^{\circ}\text{C}/-1.0\,^{\circ}\text{C}$ at least for three months in a row). Table 2 presents our classification of the years into El Niño ($SST_{anomalies} \geq +1.0\,^{\circ}\text{C}$), La Niña ($SST_{anomalies} \leq -1.0\,^{\circ}\text{C}$), and normal ($-1.0\,^{\circ}\text{C} < SST_{anomalies} < +1.0\,^{\circ}\text{C}$) years according to the ONI.

**Table 2.** El Niño, La Niña, and normal years, according to the ONI values.

| El Niño Years (NOY) | La Niña Years (NAY) | Normal Years (NOR) |
| --- | --- | --- |
| 1982 1986 1987 1991 1994 1997 2002 2009 2015 | 1988 1995 1998 1999 2007 2010 2011 2020 | 1981 1983 1984 1985 1989 1990 1992 1993 1996 2000 2001 2003 2004 2005 2006 2008 2012 2013 2014 2016 2017 2018 2019 |

The EPI's time series for each grid-cell were separated into El Niño (NOY), La Niña (NAY), and normal years (NOR) using the Oceanic Niño Index. After this process, the MWW statistical tests was applied to compare the EPI's sample in normal years, against the EPI's sample in El Niño/La Niña years. In that case, and following years classification in Table 2, the samples for performing the MWW test were:

- $EPI_{NOR} = \{EPI_{1981}, EPI_{1983}, ..., EPI_{2018}, EPI_{2019}\} \Rightarrow n_{NOR} = 23$
- $EPI_{NOY} = \{EPI_{1982}, EPI_{1986}, ..., EPI_{2009}, EPI_{2015}\} \Rightarrow n_{NOY} = 9$
- $EPI_{NAY} = \{EPI_{1988}, EPI_{1995}, ..., EPI_{2011}, EPI_{2020}\} \Rightarrow n_{NAY} = 8$

In both cases (NOR vs. NOY, and NOR vs. NAY) the sample size $N$ is "large" [49,50], so the normal approximation of MWW test was used. We obtained $p$-values maps from the MWW test, but the test says nothing about the anomaly sign. In view of this, we built maps of anomalies using the mean values in those years (i.e., $\Delta EPI_{NOY} = \overline{EPI}_{NOY} - \overline{EPI}_{NOR}$, and $\Delta EPI_{NAY} = \overline{EPI}_{NAY} - \overline{EPI}_{NOR}$).

## 5. Results

Time series of EPI in Table 1 were built over each grid-cell of the daily CHIRPSv2 in the study area, using the R library climdex.pcic [59]. Figure 4 shows the temporal behaviour of the selected EPIs over a particular grid-cell (located in Bogotá city—4°36′46″ N–74°04′14″ W). This figure depicts the NOY, NAY, and NOR as red, blue, and balck dots, respectively. In the right panel, box plots of the respective EPIs separated according to the ENSO phases are shown.

Once the EPIs time series at each grid-cell were obtained, we carried out the tests proposed in Section 4. In order to get an idea of the magnitude and spatial distribution of the EPI values over Colombia, the mean values of the normal years are depicted spatially in Figure 5. It can be seen that, in general, the spatial distribution of the EPIs has a behaviour closely tied to the spatial distribution of the mean annual precipitation shown in Figure 1b.

### 5.1. EPI's Long-Term Behavior

#### 5.1.1. Trend Test—Simple Linear Regression

Trend maps in Figure 5 (right panel) show that the spatial distribution of long-term trends differs according to the computed EPI. For example, the EPI related to dry spells (*CDD*, Figure 5f) show minimal significant grid-cells that are sparse across Colombia; however, the EPI related to wet spells (*CWD*, Figure 5g) presents a negative and significant trend over the west Colombian region, particularly on the PaR. Other EPIs show significant trends but only over relatively small areas of Colombia (e.g., *Rx1day*, *PRCP-TOT*, *R10mm*, and *R95pTOT*). Finally, *Wet days* and *SDII* show significant trends with high spatial coherence (i.e., continuous regions with significant changes) for large areas of the study area.

The EPI *Rx1day* (see Figure 5a) presents negative and significant trend located at north of the OrR, close to the boundary with Venezuela, and mainly over large areas of the CaR. The same EPI shows areas with positive trends gathered at the southern area of the PaR, near the border with Ecuador, and over the Peruvian Amazon.

The EPI *Wet days* shows significant negative trends over the whole western Colombia, while the areas with positive trends are located on the north (CaR), and on the Colombian east plains (i.e., OrR and AmR). The *SDII* shows a spatial behavior opposite to *Wet days*, since negative trends are brought together on OrR, AmR, and Car, while the positive trends are located close to the Pacific Ocean coast (PaR).

Finally, EPIs *PRCPTOT*, *R10mm*, and *R95pTOT* have similar spatial distribution of their trends. For these EPIs, there are several areas with significant positive and negative trends scattered throughout the territory. Nonetheless, the most consistent spatial distribution, and the largest trend values for these EPIs, is observed over PaR: the northern area has the greatest positive trend values; in contrast, south PaR has the most significant negative trends values.

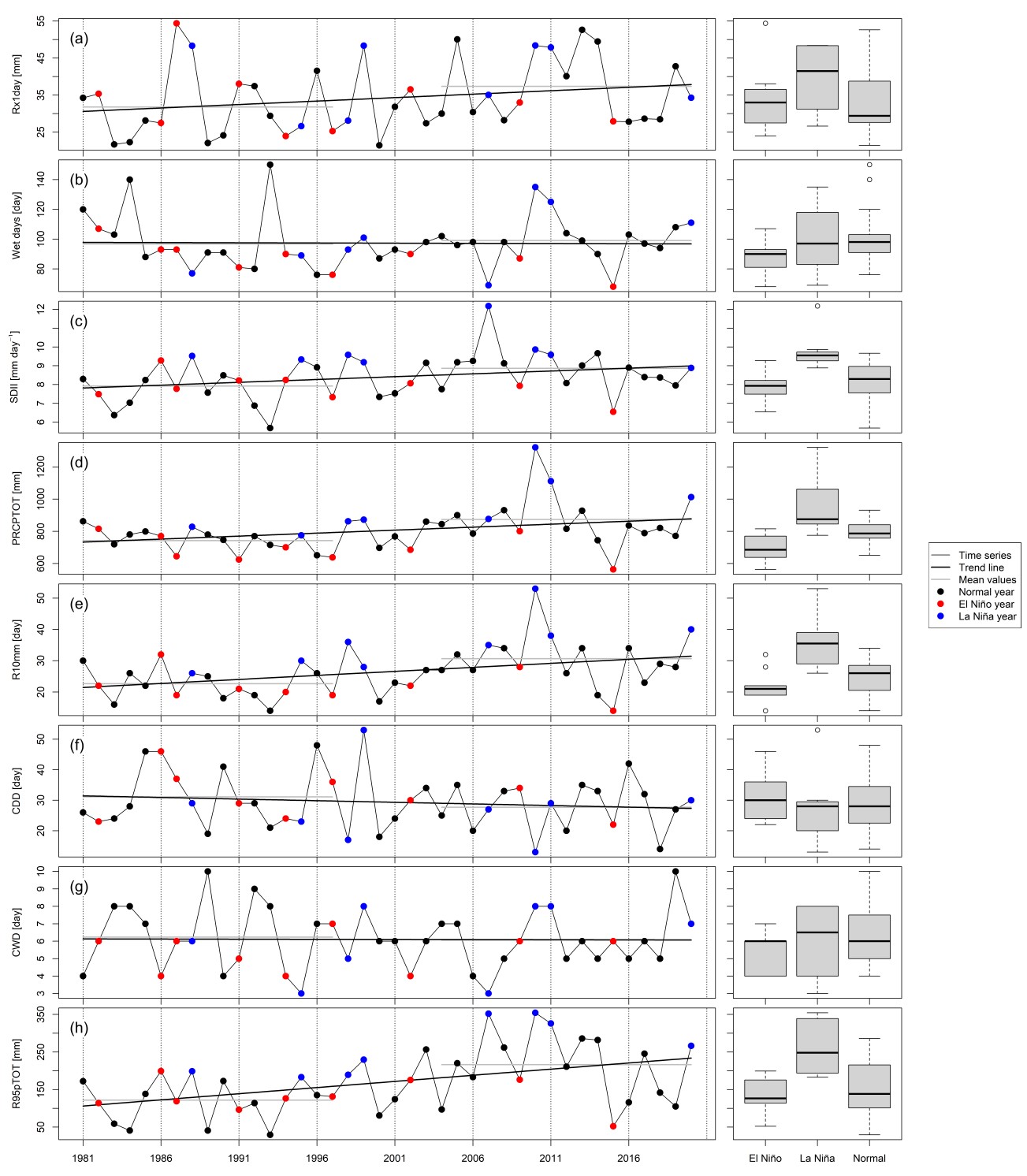

**Figure 4.** Time series of selected EPIs over a particular grid-cell (4°36′46″ N–74°04′14″ W, Bogotá city). The selected EPIs are: (**a**) *Rx1day*; (**b**) *Wet days*; (**c**) *SDII*; (**d**) *PRCPTOT*; (**e**) *R10mm*; (**f**) *CDD*; (**g**) *CWD*; and (**h**) *R95pTOT*. The left panel shows the extracted time series, where the normal (black dots; NOR), El Niño (red; NOY), and La Niña (blue; NAY) years have been identified. The black line represents the time series trend, and the horizontal gray lines show the mean values in both periods (1981–1997 and 2004–2020). According to the ENSO phase, the right panel depicts box plots of each EPI samples.

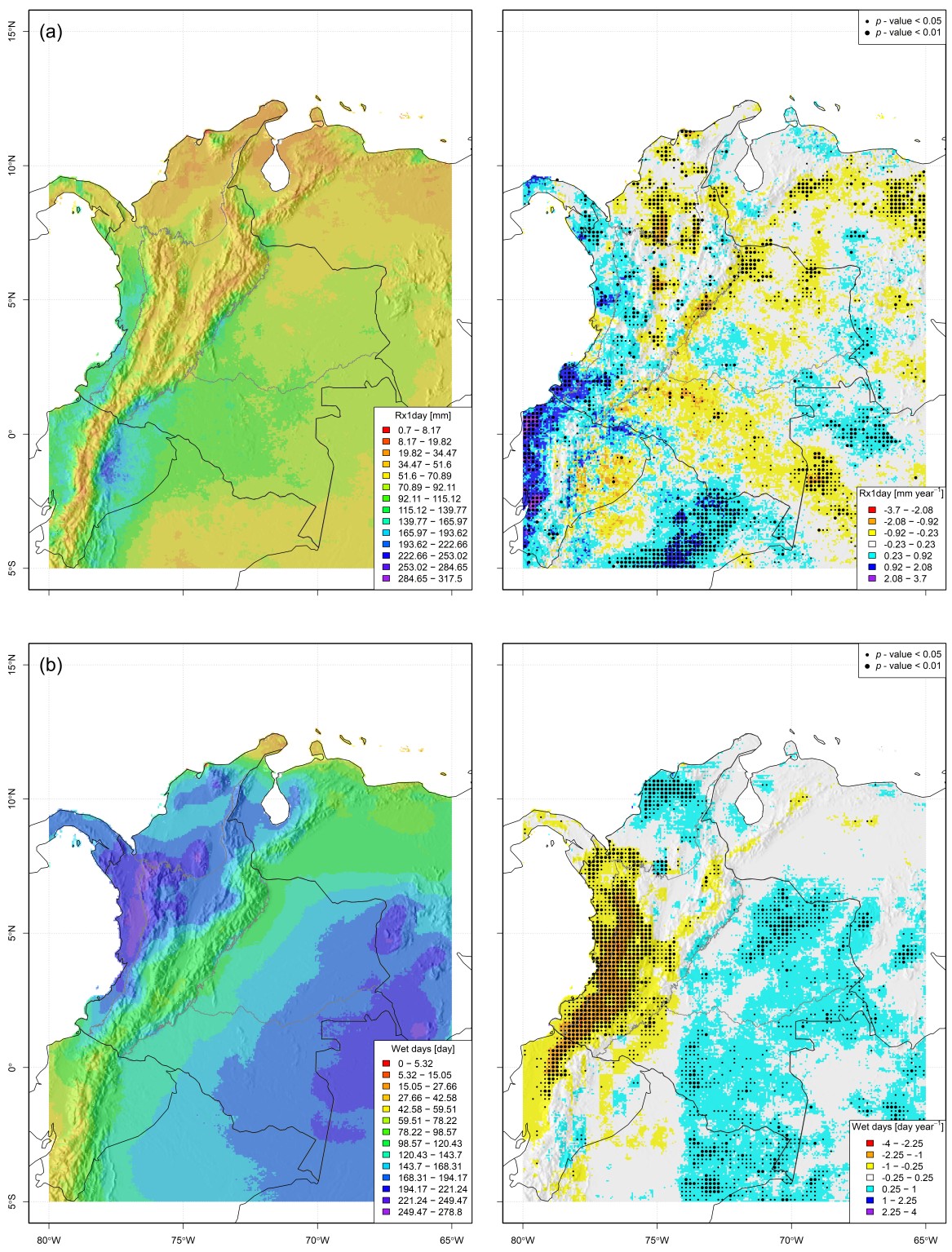

**Figure 5.** *Cont.*

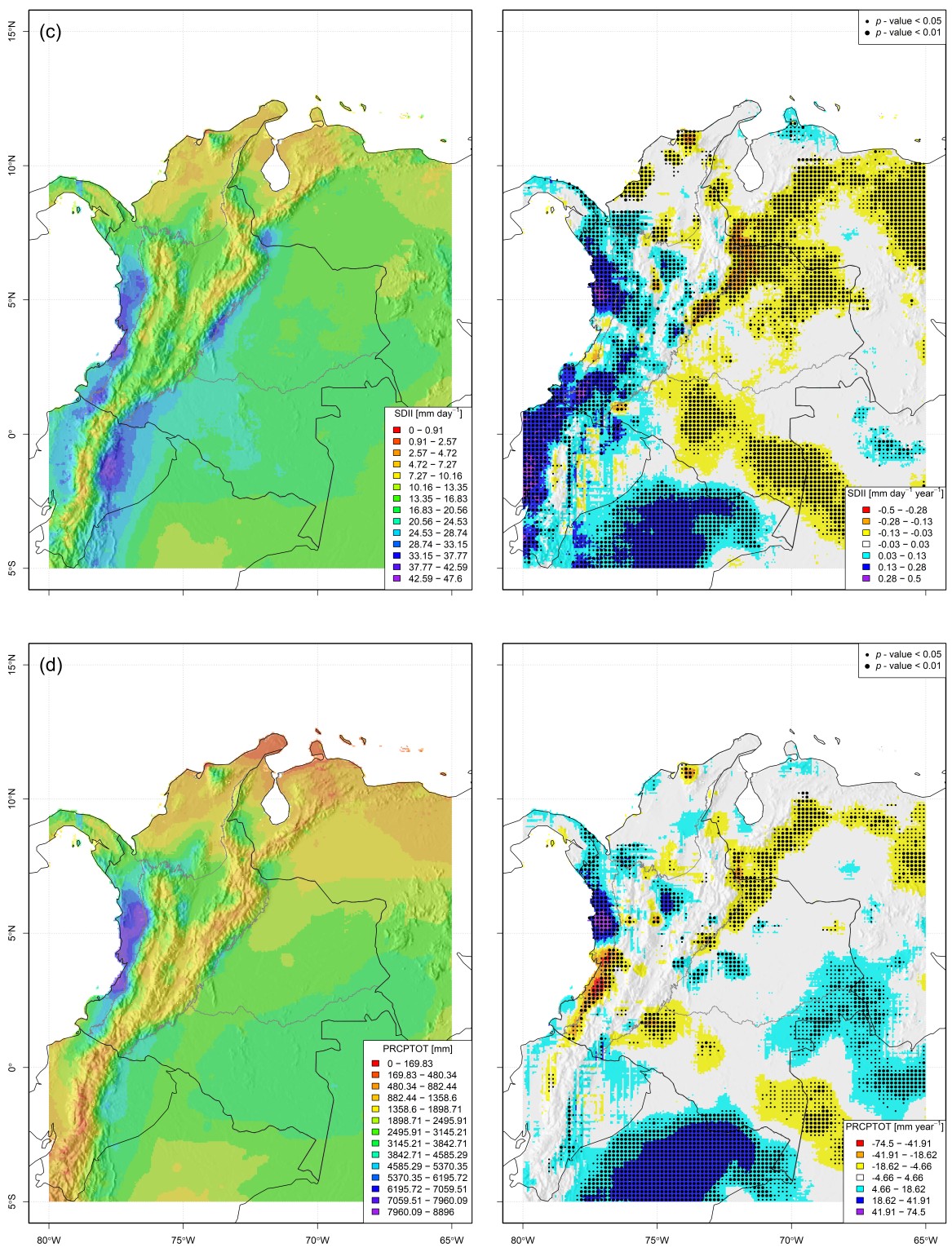

**Figure 5.** *Cont.*

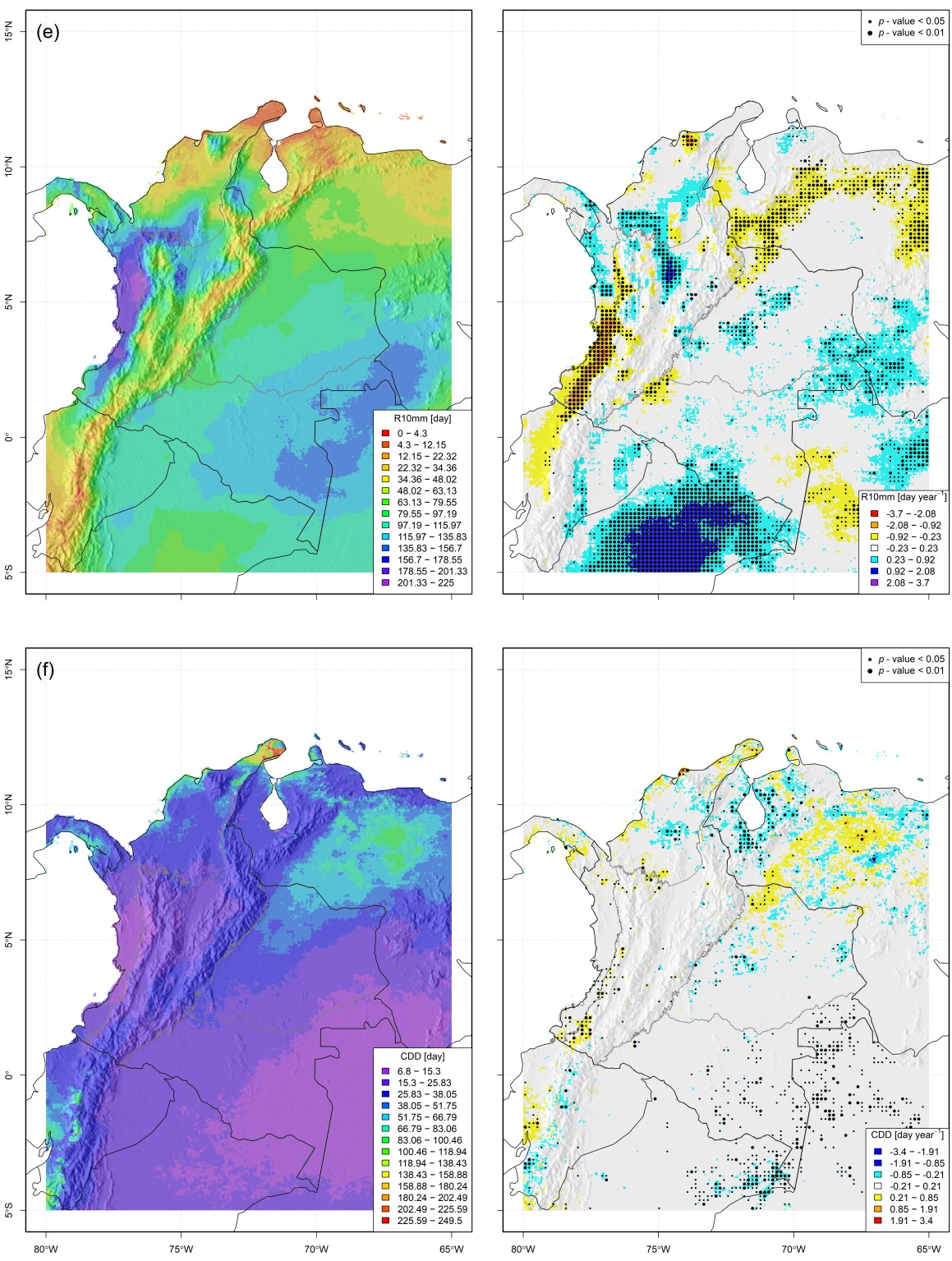

**Figure 5.** *Cont.*

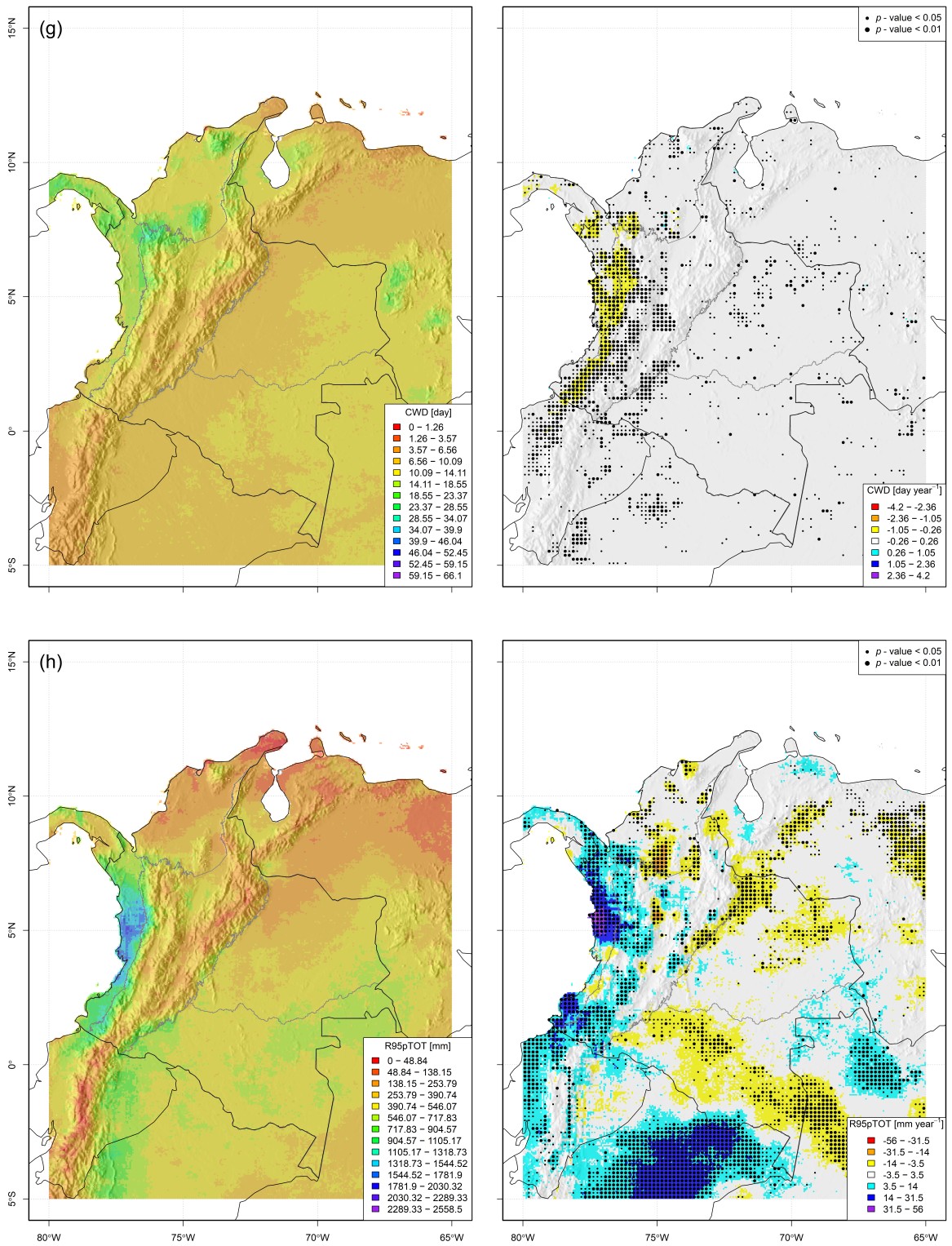

**Figure 5.** Spatially distributed mean values of (**a**) *Rx1day*, and (**b**) *Wet days* computed for the normal years (NOR; left panel). Long-term trend of the respective EPIs, as the slope of the simple linear regression *EPI* $\sim$ *t*; the black points depict the *p*-values related to the slope of the regression (right panel). The showed EPI are *SDII* (**c**), and *PCRPTOT* (**d**). The showed EPI are *R10mm* (**e**), and *CDD* (**f**). The showed EPI are *CWD* (**g**), and *R95pTOT* (**h**).

Table 3 presents the distribution of different quantiles of the selected EPIs on the study area. These results show the non-simetrical distribution of EPIs' trends in the study area.

**Table 3.** Dispersion of the slope values of the trend test in the study area.

| ID | Mean | Min | 5% | 50% | 95% | Max | Units |
|---|---|---|---|---|---|---|---|
| *Rx1day* | 0.03 | −2.17 | −0.59 | −0.01 | 0.80 | 3.34 | mm year$^{-1}$ |
| *Wet days* | 0.05 | −2.46 | −0.74 | 0.14 | 0.47 | 3.67 | day year$^{-1}$ |
| *SDII* | 0.01 | −0.31 | −0.09 | −0.01 | 0.16 | 0.46 | mm day$^{-1}$ year$^{-1}$ |
| *PRCPTOT* | 2.19 | −67.70 | −10.14 | 1.46 | 20.77 | 65.64 | mm year$^{-1}$ |
| *R10mm* | 0.07 | −2.20 | −0.37 | 0.06 | 0.71 | 3.39 | day year$^{-1}$ |
| *CDD* | −0.02 | −1.27 | −0.21 | −0.04 | 0.21 | 3.11 | day year$^{-1}$ |
| *CWD* | −0.02 | −0.63 | −0.17 | 0.00 | 0.08 | 3.82 | day year$^{-1}$ |
| *R95pTOT* | 1.08 | −21.87 | −6.63 | −0.10 | 13.70 | 50.94 | mm year$^{-1}$ |

5.1.2. Long Term Change—LTC

From the EPI time series obtained at each grid-cell, the MWW test was performed using two time series samples for two periods (1981–1997, and 2004–2020) to determine if the samples belonged to the same population. A small *p*-value indicates that samples belong to different population; however, the MWW test does not point out the change direction: i.e., if the values from the first sample are systematically higher (or lower) than those from the second sample. Therefore, to determine the direction of the change, the long-term change (LTC) was calculated using the EPI mean values difference between the periods 1981–1997 and 2004–2020 (i.e., $\Delta EPI_{LTC} = \overline{EPI}_{2004-2020} - \overline{EPI}_{1981-1997}$). As expected, the spatial distribution of EPIs long-term changes shows a spatial behavior similar to trend maps in Figure 5. The dispersion of change values $\Delta EPI$ in the study area is presented in Table 4.

The results of the assessed changes of *Rx1day*, are presented in Figure 6c. The map shows the most significant magnitude of the positive LTC on the south PaR ($\Delta Rx1day > +67$ mm), and the largest negative LTC occurs on the south of CaR, close to the boundary with the mountainous region AnR ($\Delta Rx1day < -30$ mm).

The maps for $\Delta Wet\ days$ are presented in Figure 7c. The EPI's negative and significant LTC are gathered in western Colombia (on the PaR): in a large area of this region, there were reductions of wet days with a magnitude greater than 25 days. Furthermore, adjoining the foothills of the *Cordillera Occidental de Colombia* (Colombian Western Mountain Range), the number of wet days reduced by up to 68 days. On the other hand, positive and significant LTC of this EPI occur in eastern Colombia (i.e., OrR + AmR), and the central CaR. Precisely in CaR, the largest magnitude of the positive LTC is presented.

Figure 8c depicts the maps for $\Delta SDII$. The LTC is significant and positive on PAR ($\Delta SDII > +7$ mm day$^{-1}$ in large part of the region). This EPI shows a significant negative LTC in the eastern Colombian plains (OrR+AmR); the largest magnitude of the negative change is located in the foothills of the *Cordillera Oriental de Colombia* (Colombian Eastern Mountain Range; $\Delta SDII < -3$ mm day$^{-1}$).

For *PRCPTOT*, the most important positive LTC occurs in the northernmost of PaR ($\Delta PRCPTOT > +1000$ mm); at the same time, the most important negative LTC occurs at the south of PaR ($\Delta PRCPTOT < -480$ mm). The positive and significant change observed in the Peruvian Amazon, in the center of AnR, and the easternmost of Colombia, around the triple border with Venezuela and Brazil, should be highlighted too. The maps constructed for $\Delta PRCPTOT$ are shown in Figure 9c.

The EPI *R10mm* shows that the most important positive LTC occurs in the boundary between AnR and CaR ($\Delta R10mm > 25$ mm) and on the Peruvian Amazon. The largest negative LTC for *R10mm* is located in the south of PaR, in a relatively narrow zone near the foothills of the *Cordillera Occidental*. The map depicting the LTC $\Delta R10mm$ is shown in Figure 10c.

Dry spells show significant negative LTC in the south of PaR, but the rest of the study area shows LTC spatial behavior that is not consistent (see $\Delta CDD$; Figure 11c). Wet spells

($\Delta CWD$ map; see Figure 12c) show significant negative LTC over all PaR; the magnitude of this negative change is $\Delta CWD < -23$ days in much of this region.

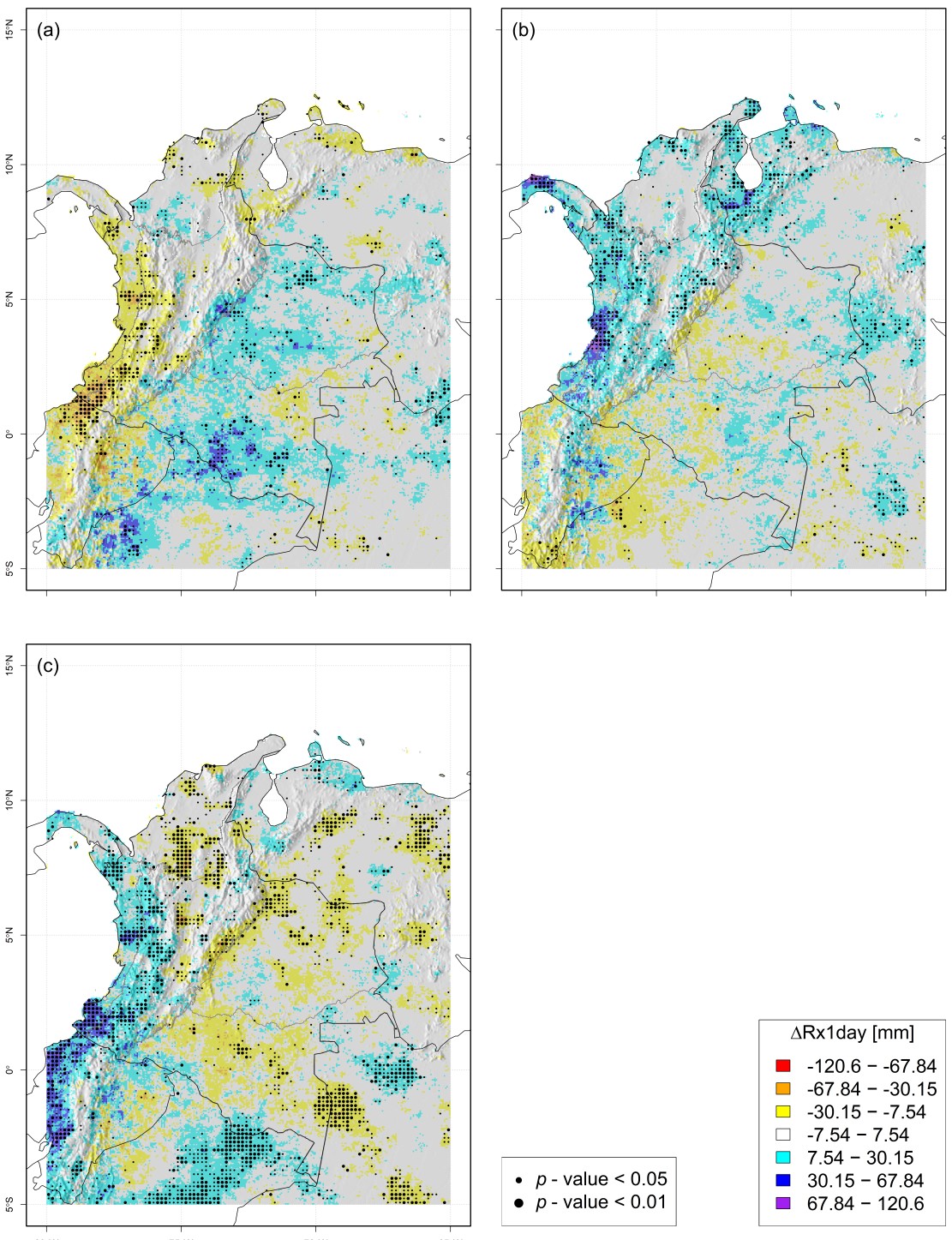

**Figure 6.** Assessed changes computed for *Rx1day*. The colours shows the difference between the EPI's mean value computed for: (**a**) El Niño years anomalies $\Delta EPI_{NOY} = \overline{EPI}_{NOY} - \overline{EPI}_{NOR}$; (**b**) La Niña years anomalies $\Delta EPI_{NAY} = \overline{EPI}_{NAY} - \overline{EPI}_{NOR}$; and (**c**) long term change $\Delta EPI_{LTC} = \overline{EPI}_{2004–2020} - \overline{EPI}_{1981–1997}$. In all maps, the black points depict the *p*-value of MWW test.

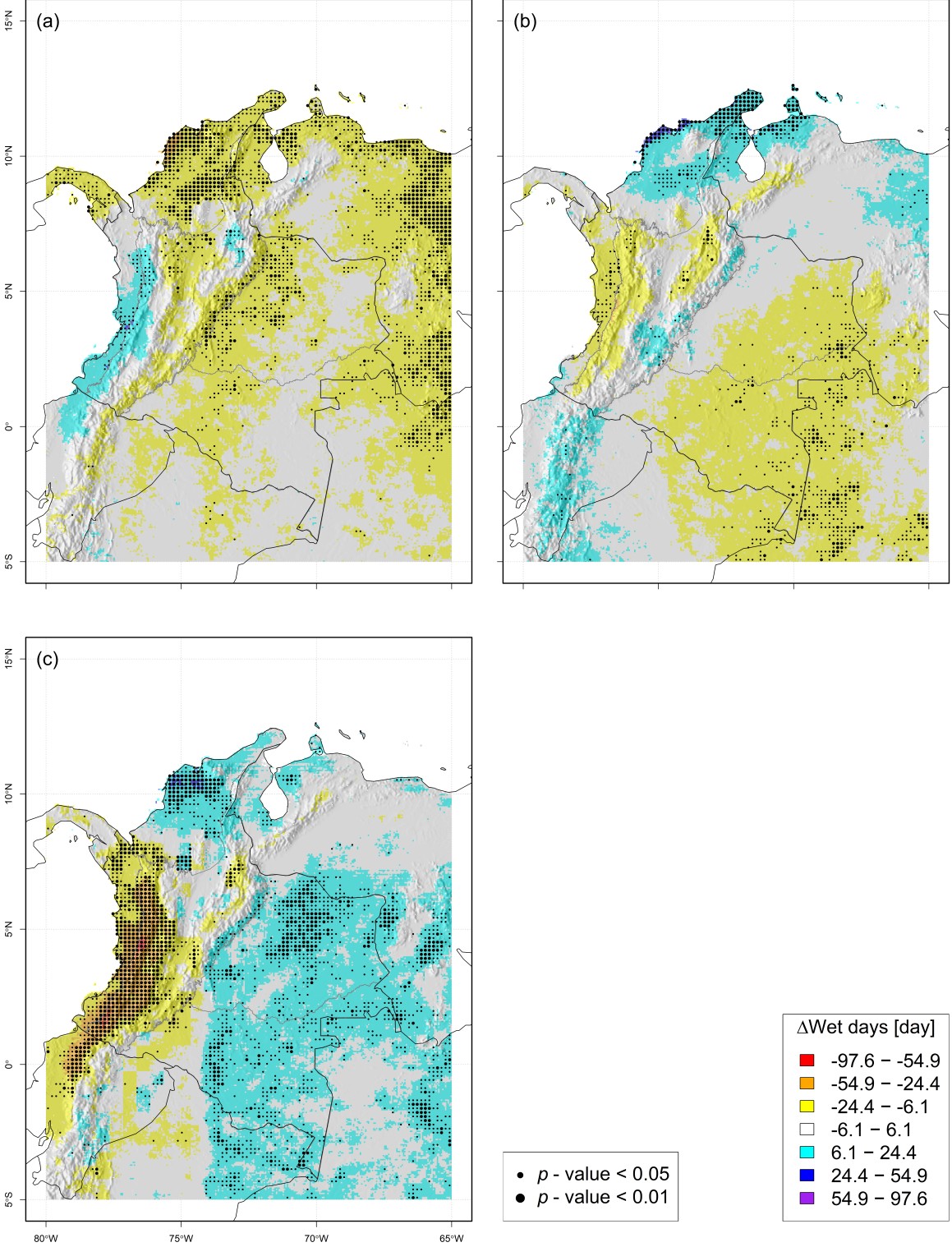

**Figure 7.** Assessed changes computed for *SDII*. The colours shows the difference between the EPI's mean value computed for: (**a**) El Niño years anomalies $\Delta EPI_{NOY} = \overline{EPI}_{NOY} - \overline{EPI}_{NOR}$; (**b**) La Niña years anomalies $\Delta EPI_{NAY} = \overline{EPI}_{NAY} - \overline{EPI}_{NOR}$; and (**c**) long term change $\Delta EPI_{LTC} = \overline{EPI}_{2004-2020} - \overline{EPI}_{1981-1997}$. In all maps, the black points depict the *p*-value of MWW test.

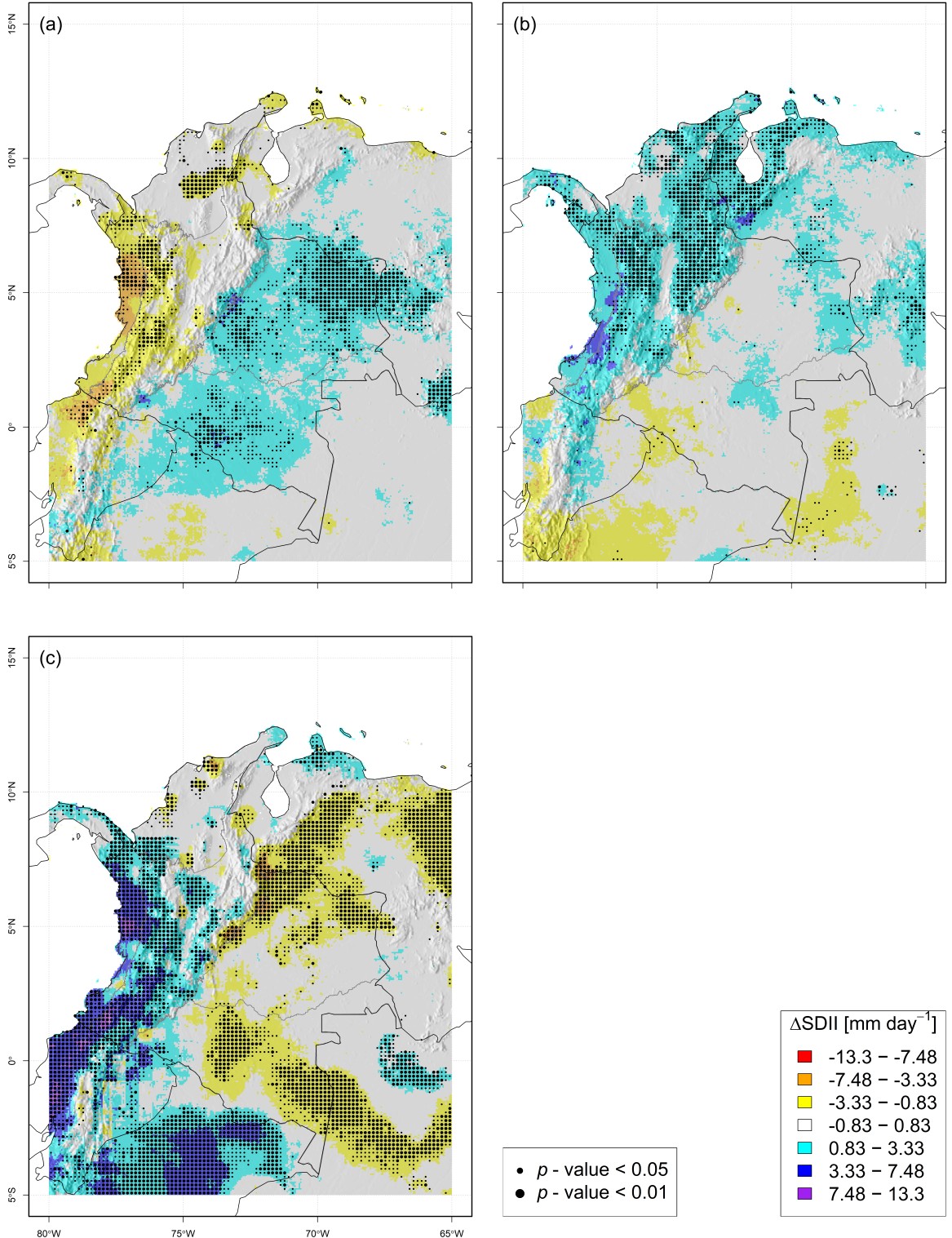

**Figure 8.** Assessed changes computed for *Wet days*. The colours shows the difference between the EPI's mean value computed for: (**a**) El Niño years anomalies $\Delta EPI_{NOY} = \overline{EPI}_{NOY} - \overline{EPI}_{NOR}$; (**b**) La Niña years anomalies $\Delta EPI_{NAY} = \overline{EPI}_{NAY} - \overline{EPI}_{NOR}$; and (**c**) long term change $\Delta EPI_{LTC} = \overline{EPI}_{2004-2020} - \overline{EPI}_{1981-1997}$. In all maps, the black points depict the *p*-value of MWW test.

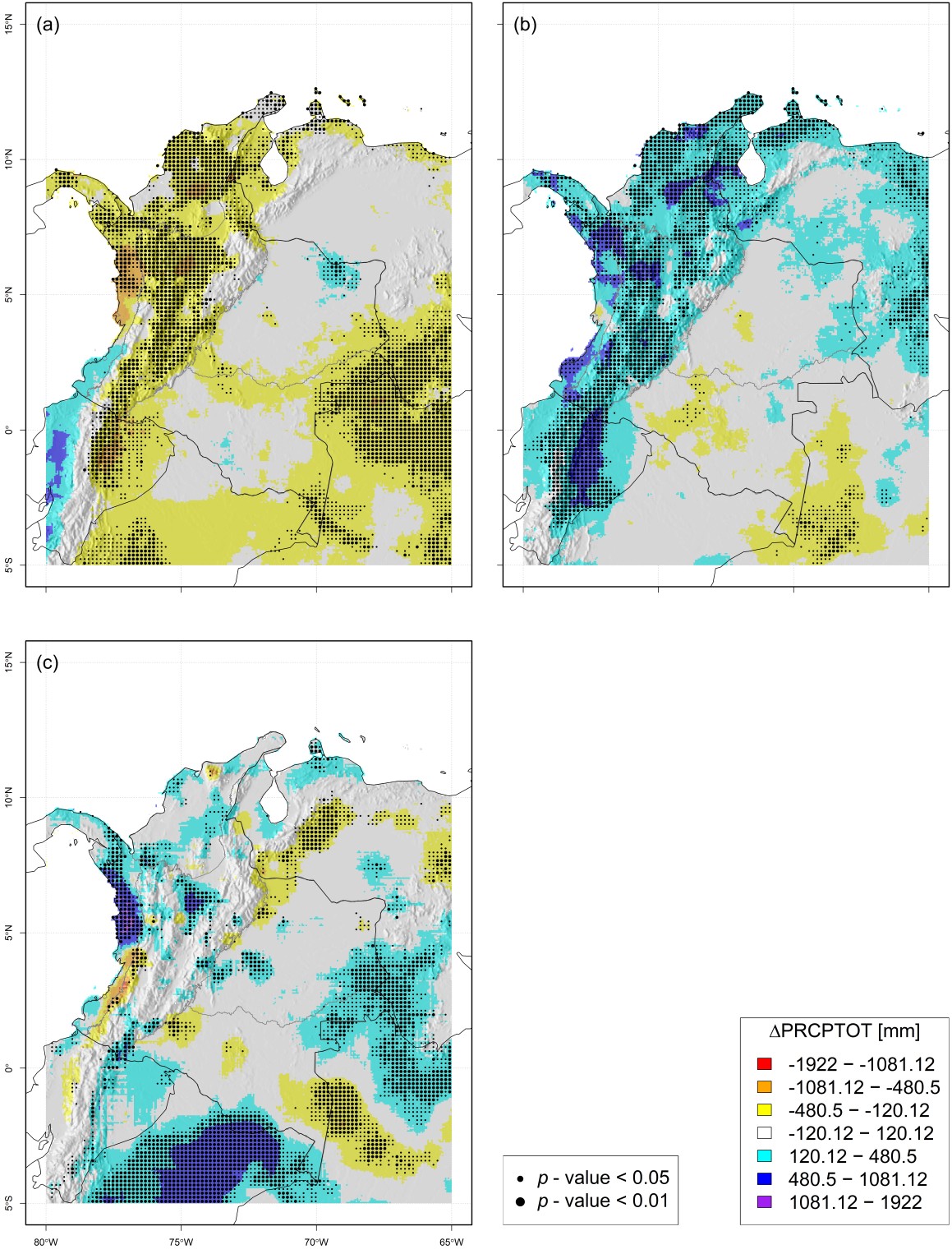

**Figure 9.** Assessed changes computed for *PCRPTOT*. The colours shows the difference between the EPI's mean value computed for: (**a**) El Niño years anomalies $\Delta EPI_{NOY} = \overline{EPI}_{NOY} - \overline{EPI}_{NOR}$; (**b**) La Niña years anomalies $\Delta EPI_{NAY} = \overline{EPI}_{NAY} - \overline{EPI}_{NOR}$; and (**c**) long term change $\Delta EPI_{LTC} = \overline{EPI}_{2004-2020} - \overline{EPI}_{1981-1997}$. In all maps, the black points depict the *p*-value of MWW test.

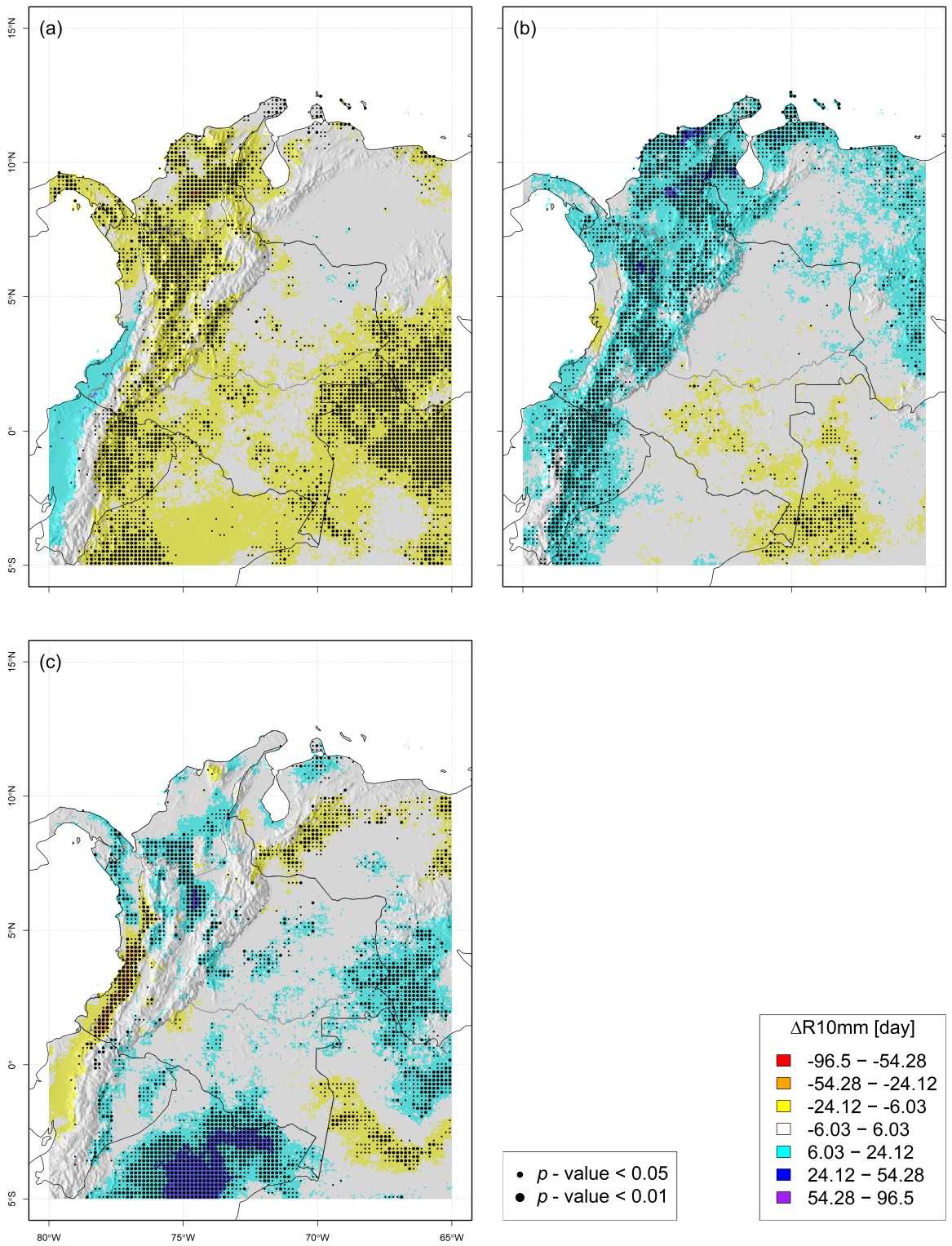

**Figure 10.** Assessed changes computed for *R10mm*. The colours shows the difference between the EPI's mean value computed for: (**a**) El Niño years anomalies $\Delta EPI_{NOY} = \overline{EPI}_{NOY} - \overline{EPI}_{NOR}$; (**b**) La Niña years anomalies $\Delta EPI_{NAY} = \overline{EPI}_{NAY} - \overline{EPI}_{NOR}$; and (**c**) long term change $\Delta EPI_{LTC} = \overline{EPI}_{2004–2020} - \overline{EPI}_{1981–1997}$. In all maps, the black points depict the *p*-value of MWW test.

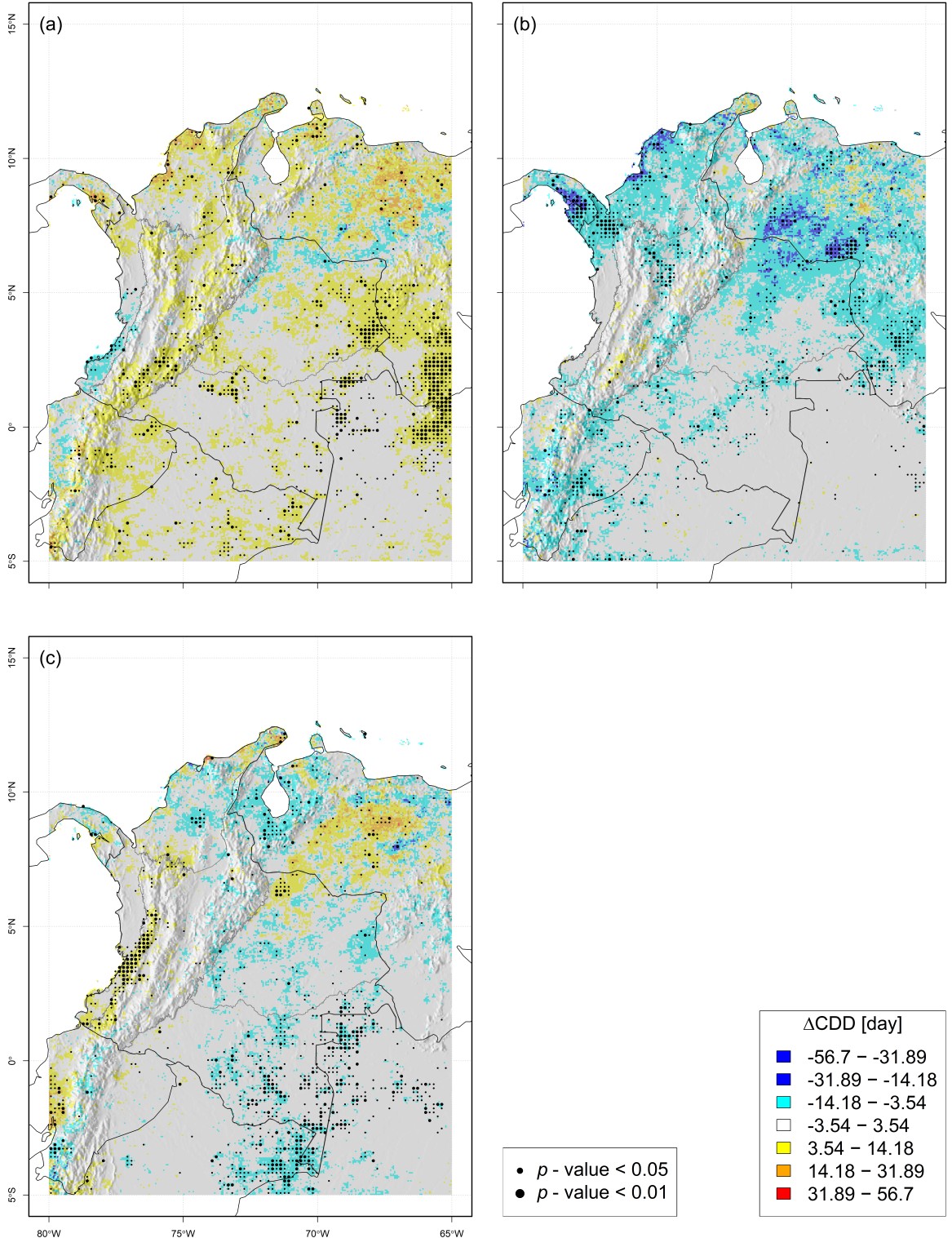

**Figure 11.** Assessed changes computed for *CDD*. The colours shows the difference between the EPI's mean value computed for: (**a**) El Niño years anomalies $\Delta EPI_{NOY} = \overline{EPI}_{NOY} - \overline{EPI}_{NOR}$; (**b**) La Niña years anomalies $\Delta EPI_{NAY} = \overline{EPI}_{NAY} - \overline{EPI}_{NOR}$; and (**c**) long term change $\Delta EPI_{LTC} = \overline{EPI}_{2004-2020} - \overline{EPI}_{1981-1997}$. In all maps, the black points depict the *p*-value of MWW test.

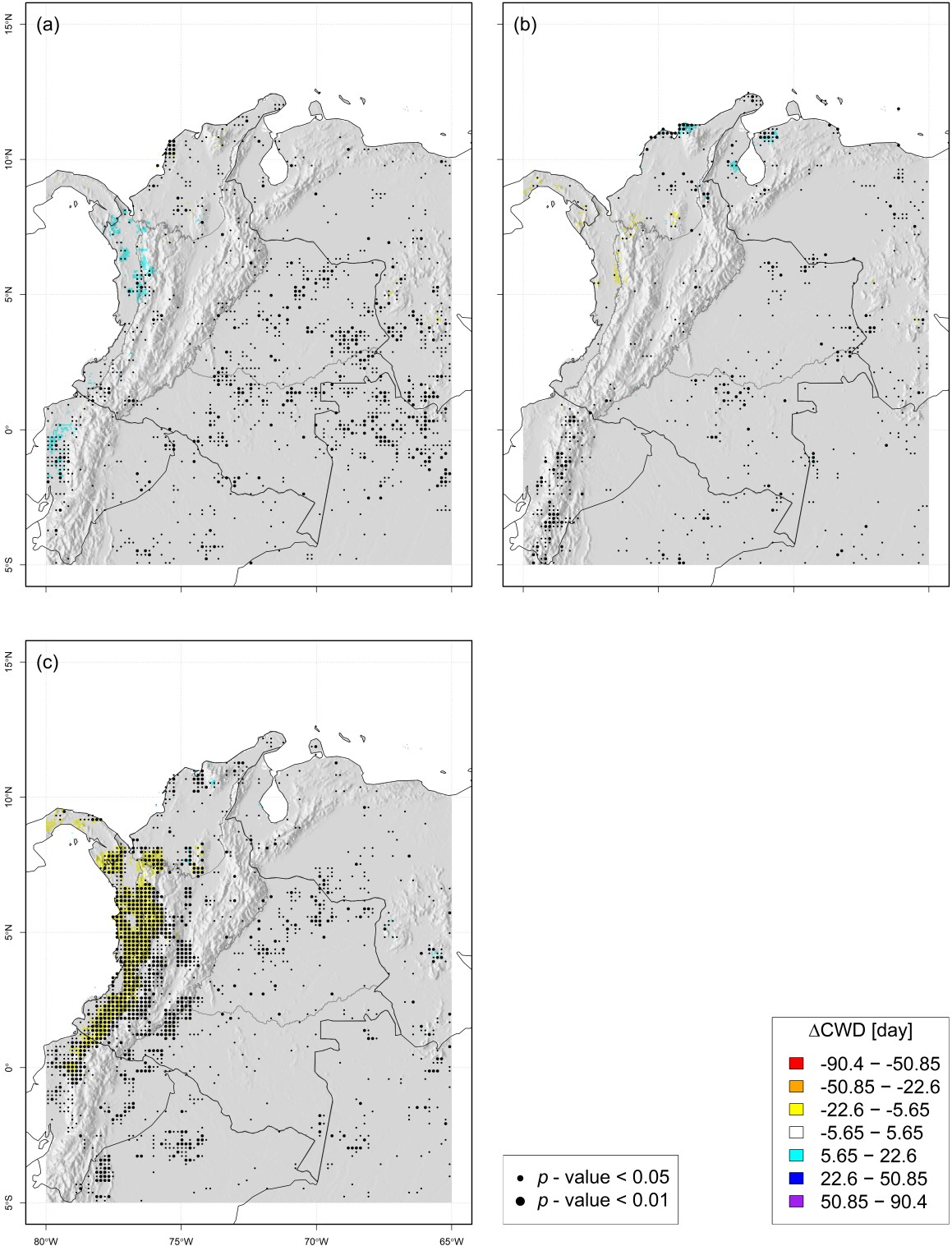

**Figure 12.** Assessed changes computed for *CWD*. The colours shows the difference between the EPI's mean value computed for: (**a**) El Niño years anomalies $\Delta EPI_{NOY} = \overline{EPI}_{NOY} - \overline{EPI}_{NOR}$; (**b**) La Niña years anomalies $\Delta EPI_{NAY} = \overline{EPI}_{NAY} - \overline{EPI}_{NOR}$; and (**c**) long term change $\Delta EPI_{LTC} = \overline{EPI}_{2004-2020} - \overline{EPI}_{1981-1997}$. In all maps, the black points depict the *p*-value of MWW test.

Finally, maps showing LTC for *R95pTOT* are presented in Figure 13c. Rainiest days of the year increased the amount of precipitation in practically all of PaR ($\Delta R95pTOT > 800$ mm). In this map, the positive LTC observed in the Peruvian Amazon should be highlighted too. The negative and significant LTC of this EPI are spatially dispersed: these changes occur in the south of CaR, and are unevenly spread in eastern Colombia (OrR + AmR).

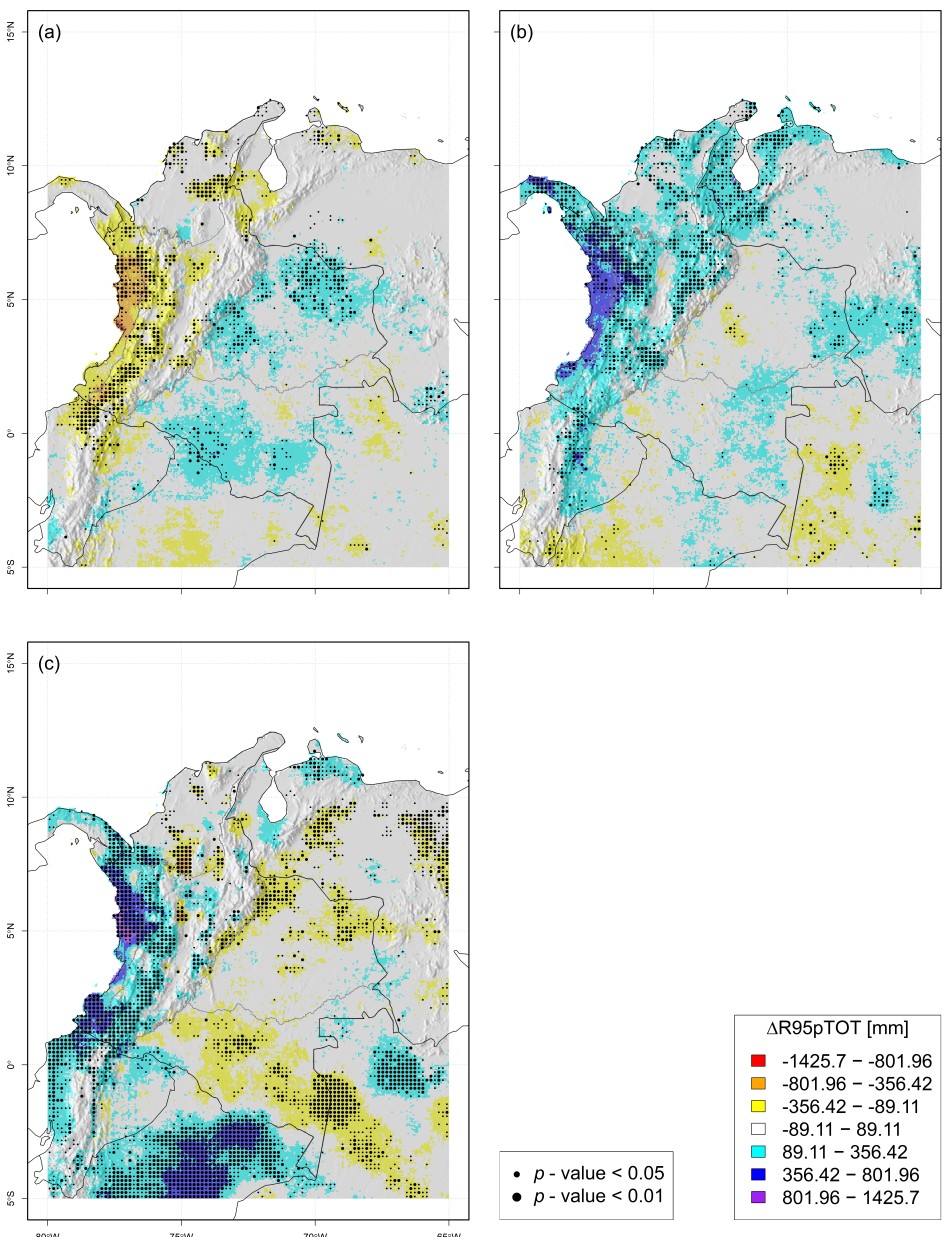

**Figure 13.** Assessed changes computed for *R95pTOT*. The colours shows the difference between the EPI's mean value computed for: (**a**) El Niño years anomalies $\Delta EPI_{NOY} = \overline{EPI}_{NOY} - \overline{EPI}_{NOR}$; (**b**) La Niña years anomalies $\Delta EPI_{NAY} = \overline{EPI}_{NAY} - \overline{EPI}_{NOR}$; and (**c**) long term change $\Delta EPI_{LTC} = \overline{EPI}_{2004–2020} - \overline{EPI}_{1981–1997}$. In all maps, the black points depict the *p*-value of MWW test.

**Table 4.** Dispersion of the values of $\Delta EPI_{LTC} = \overline{EPI}_{2004-2020} - \overline{EPI}_{1981-1997}$ in the study area.

| ID | Mean | Min | 5% | 50% | 95% | Max | Units |
|----|------|-----|-----|-----|-----|-----|-------|
| *Rx1day* | 0.83 | −56.40 | −15.31 | −0.35 | 20.67 | 90.49 | mm |
| *Wet days* | 1.71 | −68.12 | −19.88 | 4.65 | 12.71 | 88.71 | day |
| *SDII* | 0.39 | −7.05 | −2.09 | 0.01 | 4.21 | 12.06 | mm day$^{-1}$ |
| *PRCPTOT* | 85.41 | −1293.49 | −223.24 | 68.39 | 504.32 | 1747.29 | mm |
| *R10mm* | 3.04 | −52.76 | −9.29 | 3.06 | 17.82 | 87.76 | day |
| *CDD* | −0.60 | −33.18 | −5.18 | −1.12 | 5.59 | 51.59 | day |
| *CWD* | −0.52 | −17.59 | −4.59 | −0.18 | 1.88 | 82.18 | day |
| *R95pTOT* | 34.39 | −554.37 | −157.56 | 5.43 | 345.06 | 1296.08 | mm |

*5.2. EPI's Inter-Annual Variability—IAV*

The inter-annual variability of EPIs was assessed using the MWW test, which has already been used for assessed LTC in Section 5.1.2. However, in this case, the test was made between EPI values in El Niño years (NOY), or in La Niña years (NAY), against EPI values in normal years (NOR). As in Section 5.1.2, the change direction was estimated using the mean value for each sample (i.e., $\Delta EPI_{NOY} = \overline{EPI}_{NOY} - \overline{EPI}_{NOR}$, and $\Delta EPI_{NAY} = \overline{EPI}_{NAY} - \overline{EPI}_{NOR}$). The dispersion of $\Delta EPI_{NOY}$ values is presented in Table 5, while the distribution of $\Delta EPI_{NAY}$ is shown in Table 6.

**Table 5.** Dispersion of the values of $\Delta EPI_{NOY} = \overline{EPI}_{NOY} - \overline{EPI}_{NOR}$ in the study area.

| ID | Mean | Min | 5% | 50% | 95% | Max | Units |
|----|------|-----|-----|-----|-----|-----|-------|
| *Rx1day* | 1.11 | −95.49 | −16.13 | 0.62 | 19.99 | 75.14 | mm |
| *Wet days* | −5.93 | −33.89 | −15.29 | −6.10 | 4.50 | 27.77 | day |
| *SDII* | 0.15 | −7.28 | −1.82 | 0.18 | 1.92 | 5.01 | mm day$^{-1}$ |
| *PRCPTOT* | −146.38 | −1132.08 | −395.86 | −143.92 | 98.81 | 658.92 | m |
| *R10mm* | −6.15 | −29.62 | −15.88 | −6.60 | 4.01 | 26.30 | day |
| *CDD* | 2.48 | −45.25 | −3.12 | 2.14 | 8.71 | 49.42 | day |
| *CWD* | −0.47 | −17.48 | −2.62 | −0.65 | 2.43 | 11.26 | day |
| *R95pTOT* | −9.76 | −1074.27 | −163.97 | −5.26 | 144.55 | 388.83 | mm |

**Table 6.** Dispersion of the values of $\Delta EPI_{NAY} = \overline{EPI}_{NAY} - \overline{EPI}_{NOR}$ in the study area.

| ID | Mean | Min | 5% | 50% | 95% | Max | Units |
|----|------|-----|-----|-----|-----|-----|-------|
| *Rx1day* | 2.41 | −52.30 | −12.22 | 1.72 | 19.27 | 109.67 | mm |
| *Wet days* | −2.89 | −28.76 | −13.60 | −3.73 | 10.05 | 36.77 | day |
| *SDII* | 0.41 | −5.05 | −1.22 | 0.36 | 2.35 | 6.42 | mm day$^{-1}$ |
| *PRCPTOT* | 111.07 | −365.25 | −172.55 | 99.84 | 470.86 | 1439.97 | mm |
| *R10mm* | 3.34 | −21.39 | −8.36 | 3.35 | 15.45 | 35.58 | day |
| *CDD* | −2.68 | −40.79 | −10.33 | −2.02 | 2.08 | 31.92 | day |
| *CWD* | 0.00 | −10.18 | −2.32 | 0.01 | 2.27 | 13.36 | day |
| *R95pTOT* | 54.37 | −316.86 | −100.74 | 40.31 | 258.22 | 971.92 | mm |

Results from the previous analysis are in the map of Figure 6a for $\Delta Rx1day_{NOY}$, and in the map of Figure 6b for $\Delta Rx1day_{NAY}$. During NOY, negative anomalies on PaR are observed, while positive anomalies in scattered areas of the eastern Colombian plains (OrR + AmR) are shown. The EPI's values during NAY indicate positive anomalies concentrated in PaR, while the rest of the study area presents a scattered spatial behavior of these anomalies.

Maps in Figure 7a,b depict the results for $\Delta Wet\ days_{NOY}$ and $\Delta Wet\ days_{NAY}$, respectively. As seen in these maps, negative anomalies occur during NOY, mainly on CaR, and in scattered areas of OrR, close to the mountainous area. During NAY, anomalies in CaR are positive and significant over *La Guajira* peninsula (the northernmost Colombian—and South American—continental territory), and the Caribean shoreline.

Maps of $\Delta SDII_{NOY}$ and $\Delta SDII_{NAY}$ are presented in Figure 8a,b, respectively. According to what is observed there, during NOY, this EPI presents negative and significant anomalies on the central CaR, and mainly on the North of PaR. In contrast, the eastern

plains show positive anomalies. On the other hand, during NAY there are positive anomalies on CaR and AnR, while the rest of the country does not show significant anomalies for this EPI.

Maps of $\Delta PRCPTOT_{NOY}$ and $\Delta PRCPTOT_{NAY}$ are presented in Figure 9a,b, respectively. This EPI has an interesting behavior since the anomalies during NOY are opposite to those observed during NAY: i.e., $\Delta PRCPTOT_{NOY}$ presents negative and significant values on CaR, AnR, and PaR; however, over the same regions, $\Delta PRCPTOT_{NAY}$ shows positive and significant values. The eastern plains of Colombia do not present differences for this EPI. Although, it is relevant to note that negative anomalies during NOY are presented on the triple border of Colombia, Venezuela, and Brazil.

Maps in Figure 10a,b depict the results for $\Delta R10mm_{NOY}$ and $\Delta R10mm_{NAY}$, respectively. The anomalies spatial distribution for this EPI is similar to that observed for $PRCPTOT$, i.e., CaR, AnR, and PaR show negative and significant anomalies in NOY. In contrast, positive anomalies are depicted during NAY in the same regions. It makes perfect sense since the higher precipitation during wet days ($PRCPTOT$), the higher number of days with rainfall above 10 mm ($R10mm$).

Maps in Figure 11a,b depict the results for $\Delta CDD_{NOY}$ and $\Delta CDD_{NAY}$, respectively. For this EPI, the anomalies do not have high spatial coherence. Despite this, it can be said that, in general, during NOY, dry spells tend to be longer, while during NAY, dry spells tend to be shorter, than in NOR. The positive anomaly of $CDD$ during NOY is mainly gathered on the mountainous area (AnR) and the Caribbean (CaR); additionally, the positive anomaly of $CDD$ on the Venezuelan Amazon should be highlighted. During NAY, negative anomalies of $CDD$ are concentrated on the Caribbean shoreline and Orinoco river plains in Venezuela.

The EPI $CWD$ does not present significant anomalies with spatial coherence due to inter-annual variability. The map in Figure 12a shows the assessed anomalies for $\Delta CWD_{NOY}$, and the map in Figure 12b shows the deviations for $\Delta CWD_{NAY}$.

Lastly, maps in Figure 13a,b depict the results for $\Delta R95pTOT_{NOY}$ and $\Delta R95pTOT_{NAY}$, respectively. These maps show that during NOY, the values of $R95pTOT$ present significant and negative anomalies over PaR, and in other scattered areas of northern Colombia; conversely, in the eastern plains of Colombia, positive anomalies are observed (OrR + AmR), but that values have less spatial regularity than the observed in PaR. However, the anomalies during NAY are positive in the north of PaR, in the mountainous region (AnR), and various patches of CaR. Despite the notable positive anomalies observed in AnR + CaR + PaR during NAY, no significant and/or spatial coherent anomalies (neither positive nor negative) are observed in the remainder study area.

## 6. Discussion

The territory of Colombia has a different response to extreme precipitation events. The EPIs' answer depends on whether the change computation is assessed in the long-term fashion, or if it is assessed through anomalies driven by the ENSO phenomenon. Furthermore, EPIs' response during El Niño years is different from that observed during La Niña years. The Figure 14 facilitates the comparison between the long-term changes, and the inter-annual anomalies, computed for each EPI. The figure shows the distribution of the values calculated for $\Delta EPI_{LTC}$, $\Delta EPI_{NOY}$, and $\Delta EPI_{NAY}$, but restricted for each region.

In general, all Colombian regions show long-term changes of extreme precipitation indices to a greater or lesser extent. Still, not all of them are affected by the ENSO phenomenon. The most affected regions by ENSO are:

- the Pacific narrow lowlands at the west (PaR),
- the Andean mountainous region at the center (AnR), and
- the Caribbean sea northern plains (CaR).

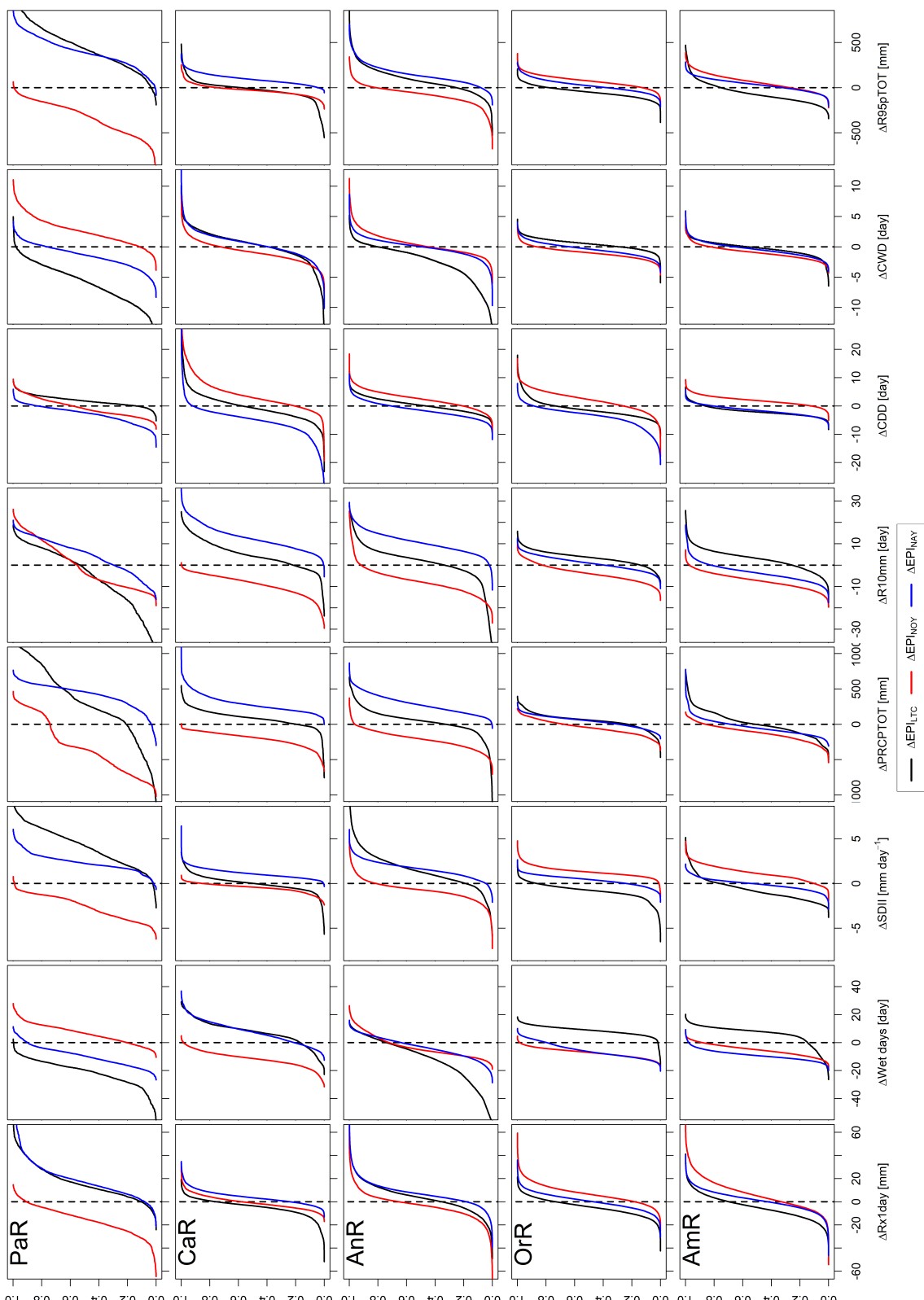

**Figure 14.** Dispersion of $\Delta EPI_{LTC}$ (black), $\Delta EPI_{NOY}$ (red), and $\Delta EPI_{NAY}$ (blue), computed for each Colombian natural regions (Pacific Region—PaR; Caribbean region—CaR; Andean region—AnR; Orinoco region—OrR; and Amazon region—AmR) .

However, it was observed that the eastern plains of Colombia (i.e., Orinoquia plus Amazonian region, OrR + AmR) have been less affected by ENSO than the rest of the

country. This observation agrees with Mesa et al. (2021; [60]), who said that ENSO affects western Colombia, while eastern regions have a precipitation behavior that is more influenced by the dynamics of the Atlantic Ocean and the Amazon Basin. The work of Salas et al. (2020; [37]) points out in the same direction; they report that during NAY, eastern Colombia (OrR + AmR) presents drier conditions, while conditions are more humid during NOY, which is an opposite behavior regarding the rest of Colombia. (AnR + CaR + PaR, western Colombia). According to [37], the synchronization of rain anomalies with the ENSO is exhibited for western Colombia, while that synchronization is not so clear for the eastern plains.

### 6.1. The Pacific Region—PaR

Regarding the Pacific region (PaR), it is observed that, in general, the EPI's long-term change repeats its trend (i.e., spatial behavior) during La Niña years. This premise is true for EPIs with positive (*Rx1day*, *SDII*, and *R95pTOT*) or negative (*Wet days*) LTC over the whole region. However, the particular spatial behavior of LTC for *PRCPTOT*, *R10mm*, and *CDD* (which have a positive change located in the northernmost area of PaR, but the rest of the region presents a negative change, or no appreciable trend), is partially repeated by the spatial pattern observed during NAY.

For NOY, several EPIs over PaR tend to present negative anomalies, which suggests less severe extreme precipitation events during these years (especially *Rx1day*, *SDII*, *PRCPTOT*, *R10mm*, and *R95pTOT*). Moreover, anomalies during NOY have a spatial behavior opposite to long-term changes. On the other hand, the EPI *Wet days* presents a slight positive and significant change over the region, which does not coincide with the observed long-term change. Finally, the EPI related to the dry spells (-*CDD*) and wet spells (*CWD*) do not present trends during NOY.

It is relevant to remark which of the changes analyzed (i.e., long-term change—LTC, or anomalies due to inter-annual variability—IAV) is the most important for the PaR. This analysis also requires considering each EPI since the importance of the change depends on the one analyzed. Thus, the long-term change was more important for *Wet days*, *SDII*, *R10mm*, *CWD*, and *R95pTOT*. In Figure 14, these indices distribution shows the greatest absolute values for $\Delta EPI_{LTC}$. Also, the LTC distributions for these EPIs are mainly positive or negative, which means spatial coherence of the assessed change. On the other hand, the difference due to inter-annual variability was more important than the long-term one for *PRCPTOT*; in Figure 14, the curves distribution are mainly negative ($\Delta EPI_{NOY}$) or positive ($\Delta EPI_{NAY}$), which means more spatial coherence than the LTC. For *Rx1day* and *CDD*, it is difficult to identify any more critical than the other.

### 6.2. The Caribbean Region—CaR

In the Caribbean region (CaR), it is observed that the climatic variability driven by ENSO produces more important shifts in the EPI than those observed in the long term (e.g., *SDII*, *PRCPTOT*, *R10mm*, *CDD*, and *R95pTOT*; see Figure 14). For these EPI, long-term changes are scattered in space, while the anomalies due to climate variability present good spatial coherence: during NOY, these EPI present negative anomalies (i.e., less severe extreme precipitation events), while during NAY, the anomalies are positive (i.e., more severe extremes events during La Niña years).

For EPIs *Rx1day* and *CWD*, both climate variability anomalies, and the long-term changes, depict scattered changes over the territory, and neither of them can be classified as the one more important. For *Rx1day*, LTC tends to be negative (especially at the south CaR), and this spatial distribution is more similar to what is observed for NOY when this EPI tends to have negative anomalies. For *CWD*, no significant changes are observed due to both long-term and climate variability.

Finally, the EPI *Wet days* deserve a detailed analysis because both the long-term change and the anomaly due to climate variability are significant in a large portion of the CaR (see Figure 14). The long-term change in this region indicates a rise in wet days,

gathered at the center of the CaR. This positive change/anomaly is mimicked during NAY, but the area affected during La Niña years is the northernmost of the Caribbean region (i.e., La Guajira peninsula).

### 6.3. The Andean Region—AnR

In the mountainous region of Colombia, long-term changes are the most important for both *Wet days* and *CWD*. For *Wet days*, there are significant negative LTC to the south of the AnR, while during NOY and NAY, there are negative anomalies that do not have as much spatial coherence as the observed long-term changes. For *CWD*, negative and significant long-term changes are also presented to the south of the AnR. Still, anomalies due to climate variability (i.e., in NOY or NAY) are not significant in this region for *CWD*.

The anomaly due to climatic variability is the most important for *PRCPTOT*, *R10mm*, and *R95pTOT* (see Figure 14). For these EPIs and throughout the region, there are significant negative anomalies during NOY, while significant positive anomalies occur during NAY. For these EPIs, the long-term change presents weak spatial coherence.

Climatic variability anomalies, and long-term changes, have a scattered spatial pattern for *Rx1day* and *CDD*. During NAY, this region presents higher values of these EPIs than during NOY; however, these climatic variability anomalies have weak spatial coherence, and positive/negative changes are, in general, not significant. Long-term changes are both spatially dispersed and not significant for both EPIs

Finally, a detailed analysis deserves the EPI *SDII*. There are significant positive changes towards the south of the AnR in the long term. Also, this EPI shows a positive anomaly during NAY, but it is gathered in the north of the AnR. In contrast, the anomaly is negative in the region's south during NOY. The EPI behavior depicts that, in general, wet days in the mountainous region of Colombia have presented rains of greater intensity due to both the long-term and the climatic variability in NAY. This observation agrees with more extreme precipitation events reported during La Niña years [28].

### 6.4. Orinoquia—OrR—and Amazon—AmR—Regions

The eastern plains region of Colombia (the Orinoquia region, plus the Amazon region; OrR + AmR) does not present widespread changes of EPIs, either due to long-term change or due to inter-annual variability. The above statement is true for *Rx1day*, *PRCPTOT*, *R10mm*, *CDD*, and *CWD*, where none of the trends analyzed (LTC or IAV) is dominant in the region, and there are no large areas with significant positive/negative changes. The changes/anomalies distributions presented in Figure 14 for these EPIs show quite similar behaviour, which confirm our previous analysis. This behavior agrees with [61], who said that the connection of the Orinoco river basin climatology with the ENSO is not clear.

The EPI *Wet days* presents an LTC that is mainly positive in this region, with significant change patches concentrated on OrR. However, this pattern of change is not replicated by any of the extreme ENSO phases (NOY or NAY; see Figure 14). The anomalies due to ENSO are mainly negative: for NOY, there are significant negative anomalies over OrR, near the mountainous region (AnR); for NAY, there are negative anomalies far away from the AnR, but they are not significant.

The EPI *SDII* presents a significant negative LTC on broad areas of the eastern Colombian plains (OrR+AmR). In the same way as the EPI *Wet days*, this spatial behavior is not replicated during any of the ENSO extreme phases: during NAY, no significant *SDII*'s anomalies were observed; however, in NOY, significant positive differences were observed throughout the territory.

Finally, *R95pTOT* exhibits LTC spatial behavior that is not replicated during extreme ENSO phases. The LTC observed for this EPI is mainly negative, with weak spatial coherence. During NAY, *R95pTOT* does not show significant anomalies. Lastly, observed positive anomalies during NOY do not have strong spatial coherence.

## 7. Conclusions

In this work, the time evolution of several extreme precipitation indices has been analyzed. The selected study area is the whole Colombian territory.

According to the results, three zones can be identified within the study domain, with observable and discernable behavior of EPIs: (i) The lowlands near the Pacific Ocean to the west (PaR); (ii) The mountainous region embedded in the middle of the country, plus the Caribbean plains to the north (AnR + CaR); and (iii) the eastern plains of the Orinoco-Amazonas basins (AmR + OrR). Long-term changes and anomalies due to inter-annual variability present different performances in these areas.

An excellent example of this disconnected behavior between regions is the EPI *PRCPTOT*:

- *PRCPTOT* does not show spatial-coherent significant changes in the mountainous area and the eastern plains in the long term. However, positive changes are observed in the northern coastal region of the Pacific Region.

- There is different behavior in the anomalies observed for this EPI during the extreme phases of ENSO in the mountainous region of Colombia (AnR) and the Caribbean (CaR). The results for this EPI coincide with the conclusions of higher (lower) moisture loads over the Colombian territory during NAY (NOY) presented by other studies [62–64].

As a result, both long-term changes and the anomalies induced by ENSO observed in EPIs indicate that they will be different according to the analyzed region. According to our study, western Colombia (AnR + CaR + PaR) has greater and more significant changes/anomalies due to LTC/IAV than eastern plains (AmR+OrR).

Colombian eastern plains (AmR + OrR) do not show a clear relationship between the EPIs' value and the IAV driven by ENSO phenomena. Therefore, to investigate the teleconnections of EPIs with other low-frequency macro-climatic phenomena is recommended. Following [60], those macro-climatic phenomena that affect the dynamics in the Atlantic Ocean, the Caribbean Sea, or the Amazon basin should be analyzed. Finally, the observed LTC in these regions were not significant too.

However, the ENSO-EPI relationship is strong in western Colombia. In general, it is observed that during NOY, there are drier conditions in the country (and, consequently, the EPI values are lower). On the opposite end, during NAY, higher moisture loads are entering mainly from the Pacific Ocean [65], leading to more severe extreme precipitation events.

Especially for the Pacific region (PaR), the sign of the long-term changes coincides with the anomalies observed during NAY (which is particularly true for *Rx1day*, *SDII*, *PRCPTOT*, and *R95pTOT*). It may appear that the EPI anomalies observed during any of the ENSO phases (in this particular case, during NAY) will be the conditions of extreme precipitation events in a future climate change scenario. It is a question that should be answered, taking into account the climate change impact on the ENSO phenomenon.

The previous question faces the unpredictability of the intensity and frequency of the extreme phases of the ENSO phenomenon in the future [25,66,67]. Answering this question would require collecting more and better information about EPIs during the ENSO extreme phases, which would allow assessing their behavior during NOY/NAY in length enough time windows.

The primary source of uncertainty in the results is the short length of the CHIRPS record. The study period (hydrological years between 1981 and 2021; forty—40—years) allowed the selection of two short time windows (seventeen—17—years each) to estimate the EPI's long-term changes. Furthermore, in the same period, only nine (9) NOY were identified, only eight (8) NAY, and twenty-three (23) NOR. These samples are very small, and the results of the MWW test can be debated, at least on the grid points where the significance of the test shows that the statistical evidence is weak.

The aforementioned problem could be partially overcome if several satellite precipitation estimates—SPE—were used. In [39], six SPE were evaluated (CHIRPS among them) over the Magdalena river basin. Computing EPI from those SPE would allow an ensemble analysis. That analysis would identify areas where the changes are significant and of the

same sign on several products. Some areas would have less uncertainty about the change because, for example, all the products agree on the change sign. In the same way, areas of great uncertainty (which may be linked to less significant changes) could be identified if several SPE do not agree on the direction and significance of the change.

Finally, the authors believe that the methodology used in this paper is suitable for monitoring extreme events anomalies driven by low-frequency macroclimatic phenomena, given its robustness even with relatively small samples. The extreme events can be maximum (i.e., the longest drought of the year, the wettest day of the year, etc.) or minimum (i.e., the minimum daily temperature for each year, the minimum recorded surface pressure, etc.) of each year. Then, annual time series of the extreme phenomenon would be built. However, it should be noted that the "hydrological year" does not generally coincide with the "calendar year".

For example, the macroclimatic phenomenon might have a very low frequency (e.g., the Pacific Decadal Oscillation [68,69]). Then, more extended periods could be considered to capture the extreme events (e.g., a value of the extreme event every five years), so that the record of the extreme event follows the natural cycle of the macroclimatic phenomenon. Then, the values for each period would be separated according to some characteristic of the macroclimatic teleconnection, which differentiates its different states. For example, in our work, we use the warm, cold and neutral phases of ENSO to classify the hydrological years.

**Author Contributions:** All authors contributed to the study's conception and methodology design. Data preparation, data analysis, and writing the original draft was done by J.D.G.-O. J.D.G.-O. guided the methodology. D.E.T.-O. and O.M.B.-V. helped in reviewing the manuscript, data preparation and analysis. J.D.G.-O. was the study's supervisor. All authors have read and agreed to the published version of the manuscript.

**Funding:** The authors express their gratitude to Pontificia Universidad Javeriana, which supported this research through the VRI15-19-1 project to promote research with foreign institutions through the mobility of professors (project ID 20095).

**Institutional Review Board Statement:** Not applicable.

**Informed Consent Statement:** Not applicable.

**Data Availability Statement:** Not applicable.

**Conflicts of Interest:** The authors declare no conflict of interest.

**Sample Availability:** Samples of the compounds are available from the authors.

## Abbreviations

The following abbreviations are used in this manuscript:

| | |
|---|---|
| AmR | Amazon region of Colombia. |
| AnR | Andean region of Colombia. |
| CaR | Caribbean region of Colombia. |
| CCI | WMO's Commission for Climatology. |
| CLIVAR | WCRP's project on Climate and Ocean Variability, Predictability and Change. |
| JCOMM | Joint Technical Commission for Oceanography and Marine Meteorology. |
| ENSO | El Niño - Southern Oscillation. |
| ETCCDI | Expert Team on Climate Change Detection and Indices. |
| | ETCCDI is a joint group between CCI, CLIVAR and JCOMM. |
| IDEAM | Instituto de Hidrología, Meteorología y Estudios Ambientales. |
| | IDEAM is the Colombian national agency focused on environmental studies. |
| NAY | La Niña years. |
| NOR | Normal years. |
| NOY | El Niño years. |

OrR      Orinoco region of Colombia.
PaR      Pacific region of Colombia.
SPE      Satellite Precipitation Estimates.
WCRP    World Climate Research Programme.
WMO    World Meteorological Organization.

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
