# Peer review of "Analysis of ENSO-Driven Variability, and Long-Term Changes, of Extreme Precipitation Indices in Colombia, Using the Satellite Rainfall Estimates CHIRPS"

_water, doi:10.3390/w14111733_

Round 1

Reviewer 1 Report

Topic of interest for the readers regarding the systematic analysis of the phenomena.

The framework and objective of the study is well defined, being expressed in the two viewpoints that concludes Section 1.

Lines 12 – 19

Several acronyms, namely related with peaks, are not defined and probably not known by the normal reader.

Line 32 – correct the term “using”.

Section 3. Data

Description could be improved regarding the traceability and the accuracy of measurement data and the instrumentation that supports those measurements.

Section 4. Methodology

The description of the selected analysis should include information about the reason to choose the approaches and tools.

Section 5.

Well described regarding the main target of the study. However, some remarks could be done regarding the interest of the methodology to be applied to other climatic phenomena with the type of variability found in this study.   

Reviewer 2 Report

Analysis of ENSO-driven variability, and long-term changes, of Extreme Precipitation Indices in Colombia, using the Satellite Rainfall Estimates CHIRPS

This study looks for compare different extreme precipitation indices over Colombia for long-term changes and considering ENSO phases.

Major comments:

It is no clear how the authors define an extreme event. They use a group of indices developed by the CCI/CLIVAR/JCOMM ETCCDI group but the do not provide any reference of these indices.

The main objetive is “analyse diverse Extreme Precipitation Indices over Colombia from two viewpoints: i) the long-term change of the EPIs and ii) the inter-annual variability of the EPIs considering the ENSO phases.” But why? What is the science question? Why is useful this analysis?

Overall the manuscript is confusing and I do not find a clear path through it.

Minor comments:

Abstract:

What do you mean by each grid-point? Which grid?

Introduction:

How do you define an “extreme event”? Only for the tails of the PDF? This is to general.

What do you mean that the results of studies of climate extremes generate uncertainty? Why? This complete paragraph is confusing. Rewrite.

Data

It would useful mention the uncertainties of the precipitation products.

Round 2

Reviewer 2 Report

No comments.